# Enforcing Hard Linear Constraints in Deep Learning Models with Decision Rules

**Gonzalo E. Constante**
Davidson School of Chemical Engineering
Purdue University
geconsta@purdue.edu

**Hao Chen**
Davidson School of Chemical Engineering
Purdue University
chen4433@purdue.edu

**Can Li**
Davidson School of Chemical Engineering
Purdue University
canli@purdue.edu

## Abstract

Deep learning models are increasingly deployed in safety-critical tasks where predictions must satisfy hard constraints, such as physical laws, fairness requirements, or safety limits. However, standard architectures lack built-in mechanisms to enforce such constraints, and existing approaches based on regularization or projection are often limited to simple constraints, computationally expensive, or lack feasibility guarantees. This paper proposes a model-agnostic framework for enforcing input-dependent linear equality and inequality constraints on neural network outputs. The architecture combines a task network trained for prediction accuracy with a safe network trained using decision rules from the stochastic and robust optimization literature to ensure feasibility across the entire input space. The final prediction is a convex combination of the two subnetworks, guaranteeing constraint satisfaction during both training and inference without iterative procedures or runtime optimization. We prove that the architecture is a universal approximator of constrained functions and derive computationally tractable formulations based on linear decision rules. Empirical results on benchmark regression tasks show that our method consistently satisfies constraints while maintaining competitive accuracy and low inference latency.

## 1 Introduction

Failing to encode safety, fairness, or physical constraints has become one of the most fundamental limitations of standard deep learning models, hindering their widespread deployment in safety-critical domains, such as energy systems, autonomous vehicles, and medical diagnostics [1, 2]. However, strictly guaranteeing constraint satisfaction on a neural network poses a significant challenge, particularly in time-sensitive settings and when the constraints are high-dimensional or input-dependent.

Popular approaches to promote constraint satisfaction use penalty-based regularization, where the constraint violations are penalized in the loss function or reward; however, these approaches, known as soft methods, are unable to provide zero constraint violation during the training and inference stages and their penalty coefficients are difficult to determine. Recent work has explored projection-based strategies either as post-hoc methods or incorporated into training, which guarantees hard constraint satisfaction. Nevertheless, these methods often face scalability and run-time limitations, as they generally require iterative procedures or solving optimization problems during inference. For detailed discussions on methods for satisfying hard constraints, the reader is referred to Section 2.

39th Conference on Neural Information Processing Systems (NeurIPS 2025).

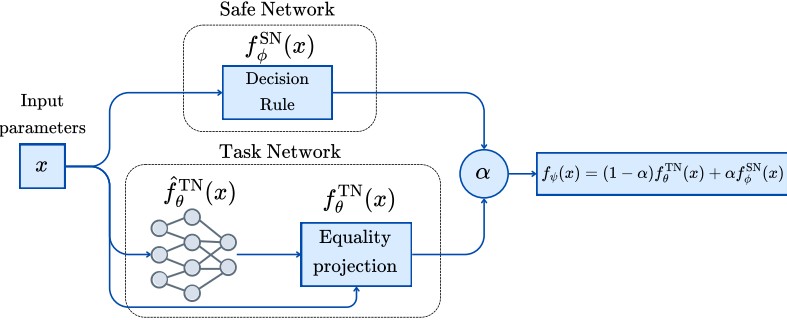

Figure 1: Proposed framework.

In this paper, we propose a framework for embedding input-output linear constraints directly into neural networks with low run-time complexity without any iterative procedure. The proposed architecture has two subnetworks: a *task network*, trained to minimize prediction loss, and a *safe network*, designed to ensure feasibility with respect to the constraint set. The proposed framework is agnostic to the task network, which is trained using stochastic gradient-descent algorithms, while the safe network is computed using a decision rule approach independent of the training of the task subnetwork. The final output is computed as a convex combination of the two subnetworks. This structure, illustrated in Figure 1, allows for seamless integration into end-to-end training pipelines and provides constraint satisfaction during training and inference stages.

The main contributions of this work are as follows:

**Framework for input-output linearly constrained predictions.** We propose a framework for embedding hard linear and input-dependent constraints directly into neural networks. The framework produces constraint-satisfying outputs in a single forward pass, and can be incorporated into any standard deep learning models.

**Theoretical guarantees and inference complexity analysis.** We provide a universal approximation theorem that characterizes the expressive power of the proposed architecture and proves that it can approximate any continuous, constraint-satisfying function arbitrarily well. Moreover, we provide a detailed analysis of the inference complexity, showing that the proposed framework introduces negligible overhead compared to the task network forward pass, particularly when exploiting constraint sparsity or when the left-hand constraint coefficients are input-independent.

**Benchmarking against state-of-the-art methods.** We compare our method to existing projection-based and penalty-based approaches on multiple tasks, showing superior constraint satisfaction, and competitive inference time and accuracy.

The remainder of the paper is organized as follows: Section 2 presents the related work on hard methods to enforce constraints. Section 3 introduces the notation and problem statement, describes the proposed framework, and presents the theoretical analysis of the proposed framework and complexity analysis. Section 4 details the training of linear decision rules. Section 5 shows the numerical experiments to evaluate the performance of the proposed methodology. Finally, Section 6 concludes the paper and outlines our future work.

## 2 Related work

**Activation functions** Simple constraints can be enforced using activation functions. For example, softmax can enforce probabilistic simplex constraints, while box constraints can be addressed using sigmoid or clipped ReLU activations followed by post-scaling to match the desired bounds.

**Projection approaches** These methods enforce constraints by projecting the output of the neural network orthogonally into the constraint set using differentiable optimization layers. In most cases, such methods require solving optimization problems during training and inference stages [3, 4, 5]. In special cases, there exist closed-form solutions for such optimization problems, represented as differentiable layers, bypassing the need to solve them during inference [6, 7]. For linear constraints, [8] propose a fast projection method for input-dependent constraints, based on a generalization of

closed-form projections for single constraints. However, their approach assumes that the number of constraints is less than the number of variables–an assumption that excludes a wide range of realistic problems in the constrained optimization literature. Other procedures involve iterative procedures, including alternating projection methods and their extensions [9, 10, 11], and unrolling differentiable gradient-based corrections [12]. However, iterative and optimization-based projection methods can be computationally expensive and slow to converge, particularly for problems with complex or large numbers of constraints.

**Sampling approaches** Inner approximation of convex constraint sets can be constructed by sampling the constraint set. A feasible point can be characterized as a convex combination of the vertices and rays obtained during the sampling step. These approaches have been proposed for homogeneous linear constraints [13] and convex polytopes [14]. However, this approach shows two limitations: (i) the number of feasible samples needed grows exponentially with the dimension of the output space, and (ii) it is restricted to constraint sets that are invariant with respect to the input.

**Preventive learning** This approach aims to enforce constraint satisfaction by shaping the training process so that the network learns to avoid infeasible regions rather than correcting outputs post hoc. [15] propose a framework that integrates constraint information into the loss function via preventive penalties and feasible region sampling, encouraging the model to produce constraint-satisfying outputs throughout training. However, these approaches rely on carefully balancing penalty terms and may still suffer from feasibility violations.

**Gauge functions** These works, which are non-iterative, are based on gauge functions that are a generalization of norms [16]. Tabas and Zhang [17, 18] propose gauge mapping, which maps a hypercube to a polytope, ensuring that the network output remains within the target polytope. One of the limiting assumptions of this method is that a feasible point of the target polytope is known. Li et al. [19] address this issue by solving an optimization problem during training and inference to find the feasible point. However, such approach is impractical for input-dependent target polytopes. For a family of convex input-independent constraints, Tordesillas et at. [20] propose RAYEN, which utilizes analytic expressions for the gauge functions, to project infeasible predictions onto the target constraint set. Recently, Liang et al. [21] proposed Homeomorphic Projection, which uses an invertible neural network to map the constraint set to a unit ball, enabling efficient projection via bisection but requiring a known interior point and adding inference overhead.

Our framework, inspired by decision rules from stochastic and robust optimization [22], generalizes the gauge mapping approach to enforce input-dependent linear constraints in neural networks through a non-iterative, closed-form correction. In contrast to existing state-of-the-art methods, it guarantees hard constraint satisfaction without requiring feasible points, iterative solvers, or specialized deep learning architectures at inference time.

## 3 Proposed framework

**Notation** We model the input space as a probability space $(\mathbb{R}^k, \mathcal{B}(\mathbb{R}^k), \mathbb{P})$, where $\mathcal{B}(\mathbb{R}^k)$ denotes the Borel $\sigma$-algebra over $\mathbb{R}^k$, and $\mathbb{P}$ is a probability measure with support $\mathcal{X} \subset \mathbb{R}^k$. The space of all Borel-measurable, square-integrable functions from $\mathbb{R}^k$ to $\mathbb{R}^n$ is denoted by $\mathcal{L}^2_{k,n}$. The space of continuous functions from $\mathcal{X}$ to $\mathbb{R}^n$ is denoted by $C(\mathcal{X}, \mathbb{R}^n)$. We denote by $\mathbb{S}^k$ the set of symmetric matrices in $\mathbb{R}^{k \times k}$. Given a matrix $A \in \mathbb{R}^{m \times n}$, $A^\top$ denotes its transpose. The identity matrix is denoted by $I$ with appropriate size inferred from the context. The standard basis vector with 1 in the first component and zeros elsewhere is denoted by $e_1 \in \mathbb{R}^k$. The notation $M \succeq 0$ denotes that matrix $M$ is symmetric positive semidefinite. All inequalities involving vectors are understood elementwise. $\mathrm{nnz}(\cdot)$ denotes the number of non-zero elements.

### 3.1 Problem setup

For a parameterized function $f_\psi : \mathcal{X} \subset \mathbb{R}^k \to \mathbb{R}^n$, we aim at enforcing the following constraint set

$$\mathcal{C}(x) := \{y \in \mathbb{R}^n \mid G(x)y = g(x), \ H(x)y \leq h(x)\}, \tag{1}$$

where $\mathcal{C}(x) \subset \mathbb{R}^n$ denotes the constraint set associated with input $x \in \mathcal{X}$. We assume that $\mathcal{C}(x)$ is a nonempty and convex set for each $x \in \mathcal{X}$. The matrices $G(x) \in \mathbb{R}^{m_{\mathrm{eq}} \times n}$ and $H(x) \in \mathbb{R}^{m_{\mathrm{ineq}} \times n}$ and vectors $g(x) \in \mathbb{R}^{m_{\mathrm{eq}}}$ and $h(x) \in \mathbb{R}^{m_{\mathrm{ineq}}}$ are linearly dependent on $x$ and define $m_{\mathrm{eq}}$ equality and $m_{\mathrm{ineq}}$ inequality constraints, respectively.

Without loss of generality, we represent (1) as

$$A(x)f_\psi(x) + s(x) = b(x), \ s(x) \geq 0, \ \forall x \in \mathcal{X}. \tag{2}$$

where $m = m_{\text{eq}} + m_{\text{ineq}}$ denotes the number of constraints, $A(x) \in \mathbb{R}^{m \times n}$ denotes the left-hand side coefficient whose first $m_{\text{eq}}$ rows correspond to the equality constraints. $b(x) \in \mathbb{R}^m$ denotes the right-hand side coefficient $b(x) = Bx$ for some $B \in \mathbb{R}^{m \times k}$, and $s(x)$ denotes the slack variable whose first $m_{\text{eq}}$ elements are equal to 0. The $\ell$th row of the left-hand side coefficient $A(x)$ is representable as $x^\top A_\ell$ for some matrix $A_\ell \in \mathbb{R}^{k \times n}$, where $\ell = 1, \ldots, m$.

## 3.2 Proposed framework

We propose an architecture composed of two subnetworks: a *safe network*, which is designed to produce outputs that satisfy the linear constraints in (2) for all inputs $x \in \mathcal{X}$, and a *task network*, which focuses on optimizing task-specific performance, such as minimizing a loss function. The outputs of these two networks are combined to form the final prediction of the overall architecture.

Specifically, given an input $x$, the final output is defined as a convex combination of the outputs of the task and safe networks:

$$f_\psi(x) = (1 - \alpha_\psi(x))f_\theta^{\text{TN}}(x) + \alpha_\psi(x)f_\phi^{\text{SN}}(x), \tag{3}$$

where $f_\theta^{\text{TN}}(x)$ and $f_\phi^{\text{SN}}(x)$ denote the outputs of the task and safe networks parameterized by $\theta$ and $\phi$, respectively. The overall parameter vector is denoted by $\psi := (\theta, \phi)$. The scalar $\alpha_\psi(x) \in [0, 1]$ is an input-dependent scalar that determines the relative contribution of the safe network to ensure feasibility of the final output.

**Task network** The goal of the task network is to provide an output that minimizes the loss function. The proposed framework is agnostic to the specific architecture of the task network, i.e., it can accommodate any neural network model and whose parameters can be learned using gradient-based methods. The output of the task network is corrected by an equality correction module reported in [6], which orthogonally projects the task network output onto the affine subspace defined by the set of equality constraints. The equality constraint satisfaction module is derived in closed-form from the KKT-conditions. The details can be found in the Appendix.

**Safe network** The role of the safe network is to provide a feasible point satisfying the equality and inequality constraints. A key challenge in enforcing input-dependent constraints is to ensure that the predicted output remains feasible for all realizations of the input space $x \in \mathcal{X}$. To this end, we propose to find the parameters of the safe network using a decision rule approach, a technique with a strong foundation in optimization under uncertainty [22]. *Decision rules*, also known as policies in the context of stochastic and robust optimization, provide a tractable approach to modeling adjustable decisions under uncertainty. A decision rule defines a functional mapping from uncertain parameters to decisions that are immune to the uncertainty set, i.e., they satisfy the constraints for all the realizations of the uncertainty set $\mathcal{X}$. Formally, if $x \in \mathcal{X}$ denotes an uncertain input drawn from a known uncertainty set, a decision rule specifies a function $y(x)$ that determines the decision corresponding to each realization of $x$. This formulation closely resembles the prediction task in supervised and reinforcement learning, where the goal is to learn an input-output mapping.

**Neural network output** The output of the task network, $f_\theta^{\text{TN}}(x)$, is projected to satisfy the equality constraints, while the output of the safe network, $f_\phi^{\text{SN}}(x)$, is constructed to satisfy both equality and inequality constraints for all $x \in \mathcal{X}$. We define the final network output as in (3). To ensure that the final output satisfies the inequality constraints, we compute the smallest value of $\alpha_\psi(x) \in [0, 1]$ such that

$$H(x)f_\psi(x) \leq h(x),$$

i.e., such that the convex combination is feasible. The equality constraints are satisfied by construction, since both $f_\theta^{\text{TN}}(x)$ and $f_\phi^{\text{SN}}(x)$ lie in the affine subspace defined by the equality constraints.

To compute $\alpha_\psi(x)$, we define the residual slack vectors:

$$s_\theta^{\text{TN}}(x) := h(x) - H(x)f_\theta^{\text{TN}}(x), \quad s_\phi^{\text{SN}}(x) := h(x) - H(x)f_\phi^{\text{SN}}(x),$$

which quantify the pointwise satisfaction margin of the inequality constraints for each network output.

Let $\mathcal{I} := \{i \in \{1, \ldots, m_{\text{ineq}}\} \mid s_{\theta,i}^{\text{TN}}(x) < 0\}$ denote the set of inequality constraints violated by the task network. For each such constraint, we determine the minimum value of $\alpha_\psi(x) \in [0, 1]$ such that the corresponding convex combination becomes feasible. The tightest such value across all violated constraints yields:

$$\alpha_\psi(x) := \max_{i \in \mathcal{I}} \frac{-s_{\theta,i}^{\text{TN}}(x)}{s_{\phi,i}^{\text{SN}}(x) - s_{\theta,i}^{\text{TN}}(x)}. \tag{4}$$

This construction ensures that the final output $f_\psi(x)$ satisfies all constraints in $\mathcal{C}(x)$, while staying as close as possible to the task network output within the feasible set. Note that although $\alpha_\psi(x)$ is computed from the slack vectors $s_\theta^{\text{TN}}(x)$ and $s_\phi^{\text{SN}}(x)$, which depend on the outputs of the task and safe networks parameterized by $\theta$ and $\phi$, respectively, it is evaluated using a `max` operation that is subdifferentiable almost everywhere. This allows gradients to propagate only through the constraint index that achieves the maximum, while all other terms are treated as constant.

To make this explicit, consider the loss $\mathcal{L}(f_\psi(x), y)$. By the chain rule,

$$\frac{\partial \mathcal{L}}{\partial \theta} = \frac{\partial \mathcal{L}}{\partial f_\psi(x)} \cdot \left( \frac{\partial f_\psi(x)}{\partial f_\theta^{\text{TN}}(x)} \cdot \frac{\partial f_\theta^{\text{TN}}(x)}{\partial \theta} + \frac{\partial f_\psi(x)}{\partial \alpha_\psi(x)} \cdot \frac{\partial \alpha_\psi(x)}{\partial \theta} \right).$$

In this expression, the term $\partial \alpha_\psi(x)/\partial \theta$ is nonzero only for the constraint selected by the `max`, leading to sparse gradients with respect to $\theta$. As a result, the effective update to the task network is modulated by both the selected constraint and the value of $\alpha_\psi(x)$, reflecting the degree of feasibility violation in the task output.

### 3.3 Computing the safe network

The safe network is computed offline, separately from the training of the task network. We formulate the computation of the safe network as the following optimization problem:

$$\begin{aligned} \underset{f_\phi^{\text{SN}} \in \mathcal{L}_{k,n}^2, s \in \mathcal{L}_{k,m}^2, t}{\text{maximize}} \quad & t \\ \text{subject to} \quad & \left. \begin{aligned} A(x) f_\phi^{\text{SN}}(x) + s(x) &= b(x) \\ s(x) &\geq t \end{aligned} \right\} \; \mathbb{P}\text{-a.s.} \end{aligned} \tag{5}$$

where the slack variable $s(x)$ is introduced to express inequality constraints in standard form. The objective in (5) encourages the safe network output to lie deep in the interior of the feasible region. The larger the slacks in the safe output, the more flexibility we have to interpolate while preserving feasibility. As a result, the final output remains closer to the task network prediction, reducing conservativeness. Note that problem (5) is an infinite-dimensional optimization problem since the variables, $f_\phi^{\text{SN}}$ and $s$, live in function spaces and the constraints must hold for all $x \in \mathcal{X}$ almost surely.

While problem (5) guarantees feasibility, it is intractable in general since finding its optimal value is known to be #P-hard [23]. To obtain a tractable surrogate, we can restrict the functional form of the decision rules [22]. Prior efforts include forms of decision rules to be affine [24], segregated affine [25], piecewise affine [26, 27], and trigonometric polynomial [28] functions. In Section 4, we provide computationally tractable inner approximations of problem (5), which can be efficiently solved using state-of-the-art constrained optimization solvers.

### 3.4 Theoretical guarantees

We prove that, under mild conditions on the task and safe network function classes, the proposed architecture is a universal approximator of continuous functions that satisfy input-dependent linear constraints.

**Assumption 1** (Task Network Function Class). *Let $\mathcal{X} \subset \mathbb{R}^k$ be compact. The task network function class $\mathcal{F}_{\text{task}} \subseteq C(\mathcal{X}, \mathbb{R}^n)$ satisfies the universal approximation property: for any continuous function $f^* : \mathcal{X} \to \mathbb{R}^n$ and any $\varepsilon > 0$, there exists a function $f_\theta^{\text{TN}} \in \mathcal{F}_{\text{task}}$ such that*

$$\sup_{x \in \mathcal{X}} \|f_\theta^{\text{TN}}(x) - f^*(x)\| < \varepsilon.$$

**Assumption 2** (Safe Network Function Class). *Let $\mathcal{C}(x) \subset \mathbb{R}^n$ be a convex constraint set for each $x \in \mathcal{X}$. The safe network function class $\mathcal{F}_{\text{safe}} \subseteq C(\mathcal{X}, \mathbb{R}^n)$ is such that for all $f_\phi^{\text{SN}} \in \mathcal{F}_{\text{safe}}$,*

$$f_\phi^{\text{SN}}(x) \in \mathcal{C}(x), \quad \forall x \in \mathcal{X}.$$

*Moreover, for every $x \in \mathcal{X}$, the convex hull of the set $\{f_\phi^{\text{SN}}(x) \mid f_\phi^{\text{SN}} \in \mathcal{F}_{\text{safe}}\}$ is dense in $\mathcal{C}(x)$.*

**Theorem 1** (Universal Approximation with Constrained Output). *Let $\mathcal{X} \subset \mathbb{R}^k$ be compact, and let $f^* : \mathcal{X} \to \mathbb{R}^n$ be a continuous function such that $f^*(x) \in \mathcal{C}(x)$ for all $x \in \mathcal{X}$, where $\mathcal{C}(x) \subset \mathbb{R}^n$ is a convex, input-dependent constraint set. Suppose Assumptions 1 and 2 hold.*

*Then, for any $\varepsilon > 0$, there exist functions $f_\theta^{\text{TN}} \in \mathcal{F}_{\text{task}}$, $f_\phi^{\text{SN}} \in \mathcal{F}_{\text{safe}}$, and a scalar $\alpha \in [0, 1]$ such that the convex combination*

$$f_\psi(x) = (1 - \alpha) f_\theta^{\text{TN}}(x) + \alpha f_\phi^{\text{SN}}(x)$$

*satisfies $f_\psi(x) \in \mathcal{C}(x)$ for all $x \in \mathcal{X}$, and*

$$\sup_{x \in \mathcal{X}} \| f_\psi(x) - f^*(x) \| < \varepsilon.$$

### 3.5 Inference complexity

**Equality Projection of Task Network Output**   When $G(x)$ is input-independent, the projection can be efficiently implemented by exploiting the closed-form structure in (16) without explicitly forming the dense matrix $I - \bar{G}G$, where $\bar{G} := G^\top (GG^\top)^{-1}$ denotes the left pseudo-inverse of $G \in \mathbb{R}^{m_{\text{eq}} \times n}$, used to project onto the affine set defined by the equality constraints. The projection is computed as $x - \bar{G}(Gx) + \bar{G}g$, requiring two matrix-vector products with $G$ and $\bar{G}$, both costing $\mathcal{O}(n \, m_{\text{eq}})$, and lightweight vector additions at $\mathcal{O}(n)$. Thus, the overall inference complexity is $\mathcal{O}(n \, m_{\text{eq}})$, regardless of the density of $\bar{G}$. When $G(x)$ is input-dependent, the projection must be recomputed for each input. In the dense case, the dominant costs are forming $G(x)G(x)^\top$ with $\mathcal{O}(n \, m_{\text{eq}}^2)$ and inverting it with $\mathcal{O}(m_{\text{eq}}^3)$, resulting in an overall complexity of $\mathcal{O}(n \, m_{\text{eq}}^2 + m_{\text{eq}}^3)$. In the sparse case, matrix-vector products cost $\mathcal{O}(\text{nnz}(G(x)))$, while forming $G(x)G(x)^\top$ costs $\mathcal{O}(\text{nnz}(G(x)) \, m_{\text{eq}})$. The inversion remains $\mathcal{O}(m_{\text{eq}}^3)$, leading to a total cost of $\mathcal{O}(\text{nnz}(G(x)) \, m_{\text{eq}} + m_{\text{eq}}^3 + n)$.

**Safe Network, Slack, and Convex Combination Complexity**   The safe network forward pass consists of a dense matrix-vector product $Fx$, resulting in a computational cost of $\mathcal{O}(n \, k)$. The slack computation, defined as $s(x) = h(x) - H(x)f(x)$, requires computing $H(x)f(x)$. In the dense case, this operation incurs a cost of $\mathcal{O}(m_{\text{ineq}} \, n)$, while in the sparse case, it reduces to $\mathcal{O}(\text{nnz}(H(x)))$, depending on the sparsity of $H(x)$. The calculation of the blending parameter $\alpha$ involves evaluating $m_{\text{ineq}}$ scalar inequalities and thus has a cost of $\mathcal{O}(m_{\text{ineq}})$. Finally, the convex combination of the task and safe network outputs requires $\mathcal{O}(n)$ operations.

**Total Inference Complexity**   Overall, the proposed framework achieves low inference complexity, summarized in Table 1. The dominant cost arises from the task network forward pass, denoted by $C_{\text{TN}}$, which refers to the computational complexity of a single forward pass through the task network. The remaining components introduce negligible overhead, especially when exploiting sparsity in the constraint matrices.

Table 1: Total inference complexity of the proposed framework. $\text{nnz}(\cdot)$ denotes the number of non-zero elements.

| Constraints | Coefficients $G(x)$ | Total Complexity |
|---|---|---|
| Dense | Input-dependent | $\mathcal{O}\left(C_{\text{TN}} + n \, m_{\text{eq}}^2 + m_{\text{eq}}^3 + n \, k + m_{\text{ineq}} \, n\right)$ |
|  | Input-independent | $\mathcal{O}\left(C_{\text{TN}} + n \, m_{\text{eq}} + n \, k + m_{\text{ineq}} \, n\right)$ |
| Sparse | Input-dependent | $\mathcal{O}\left(C_{\text{TN}} + \text{nnz}(G(x)) \, m_{\text{eq}} + m_{\text{eq}}^3 + n \, k + \text{nnz}(H(x)) + n\right)$ |
|  | Input-independent | $\mathcal{O}\left(C_{\text{TN}} + n \, m_{\text{eq}} + n \, k + \text{nnz}(H) + n\right)$ |

# 4 Offline computation of safe networks with decision rules

## 4.1 Linear decision rules for linear constraints

To reduce the computational tractability of problem (5), we restrict the functional form of the safe network to linear functions of the input, as follows:

$$f_\phi^{\text{SN}}(x) = Fx, \tag{6}$$

where $F \in \mathbb{R}^{n \times k}$. Additionally, we assume that the input space $\mathcal{X} \subset \mathbb{R}^k$ to be a nonempty, compact set defined as:

$$\mathcal{X} := \left\{ x \in \mathbb{R}^k \mid e_1^\top x = 1, \; x^\top P_j x \geq 0, \; j = 1, \dots, l \right\}, \tag{7}$$

for some symmetric matrices $P_j \in \mathbb{S}^k$. Note that set $\mathcal{X}$ corresponds to the intersection of $l$ ellipsoids intersected with an affine hyperplane constraint $e_1^\top x = 1$. For the sake of notational simplicity and without loss of generality, we assume that the first component of any $x \in \mathcal{X}$ is equal to 1. Furthermore, we assume that the linear hull of $\mathcal{X}$ spans $\mathbb{R}^k$. Under these assumptions, the slack variable $s(x)$ becomes a quadratic function of the input $x$ and problem (5) is approximated as:

$$
\begin{aligned}
\underset{F,S,t}{\text{maximize}} \quad & t \\
\text{subject to} \quad & F \in \mathbb{R}^{n \times k}, S = (S_1, \dots, S_m) \in \left( \mathbb{S}^k \right)^m, t \in \mathbb{R} \\
& \left. \begin{aligned} x^\top A_\ell F x + x^\top S_\ell x &= b_\ell^\top x, \; \ell = 1, \dots, m \\ x^\top S_\ell x &\geq t, \; \ell = m_{\text{eq}} + 1, \dots, m. \end{aligned} \right\} \; \mathbb{P}\text{-a.s.,}
\end{aligned}
\tag{8}
$$

We note that problem (8) is a semi-infinite problem due to the finite number of variables, but an infinite number of quadratic constraints. To make the above problem tractable, we first handle the equality constraints. Since all constraints are continuous in $x$, we can symmetrize and rewrite the $\ell$th quadratic equality constraint as follows:

$$x^\top H_\ell x = 0, \; \forall x \in \mathcal{X}, \tag{9}$$

where $H_\ell := \frac{1}{2} \left( A_\ell F + F^\top A_\ell^\top - e_1 b_\ell^\top - b_\ell e_1^\top \right) + S_\ell$ is a symmetric matrix.

Since the constraint is quadratic and homogeneous in $\mathcal{X}$, it equivalently holds over the conic hull of $\mathcal{X}$, denoted $\text{cone}(\mathcal{X})$. Given that $\mathcal{X}$ spans $\mathbb{R}^k$ by assumption, $\text{cone}(\mathcal{X})$ has a non-empty interior. The Hessian of the mapping $x \mapsto x^\top H_\ell x$ is given by $2H_\ell$. Consequently, if the quadratic form vanishes over the interior of $\text{cone}(\mathcal{X})$, it follows that the Hessian must be $H_\ell = 0$ [29]. Thus, the original semi-infinite equality constraint admits the equivalent finite-dimensional representation $H_\ell = 0$. This reformulation substantially enhances tractability, as it replaces the intractable semi-infinite constraint with a linear matrix equality.

Next, we can use Proposition 1 to approximate the semi-infinite constraints in (8), simplifying it into the following semidefinite program (SDP):

$$
\begin{aligned}
\underset{F,S,t}{\text{maximize}} \quad & t \\
\text{subject to} \quad & F \in \mathbb{R}^{n \times k}, S = (S_1, \dots, S_m) \in \left( \mathbb{S}^k \right)^m, \Lambda \in \mathbb{R}^{m \times l}, t \in \mathbb{R} \\
& \frac{1}{2} \left( A_\ell F + F^\top A_\ell^\top \right) + S_\ell = \frac{1}{2} \left( e_1 b_\ell^\top + b_\ell e_1^\top \right), \; \forall \ell = 1, \dots, m \\
& S_\ell - \sum_{j=1}^{l} \Lambda_{\ell j} P_j - tI \succeq 0, \; \forall \ell = m_{\text{eq}} + 1, \dots, m \\
& \Lambda \geq 0.
\end{aligned}
\tag{10}
$$

The above SDP constitutes an *inner approximation* of the original problem in (8) in the general case. This means that the feasible set of the SDP is contained within that of the original problem, ensuring tractability at the cost of some conservatism. However, when $l = 1$, this approximation becomes *exact*, and the SDP formulation recovers the original feasible set without loss of generality. This property follows from classical results on the tightness of SDP relaxations for single quadratic constraints. A key advantage of problem (10) is that it yields a SDP whose size grows polynomially with the dimensions of the input and output spaces, as well as with the number of constraints defining $\mathcal{X}$. This ensures that the problem can be solved efficiently using state-of-the-art interior-point methods.

## 4.2 Linear decision rules for jointly linear constraints

A special case of the previously presented framework arises when the left-hand side matrix is input-independent, i.e., $A(x) \equiv A$, while the right-hand side remains input-dependent as $b(x) = Bx$, where $B \in \mathbb{R}^{m \times k}$ is fixed. Under this assumption, the constraints of problem (5) reduce to the following jointly linear form:

$$A f_\phi^{\mathrm{SN}}(x) + s(x) = Bx, \ s(x) \geq 0, \ \forall x \in \mathcal{X}. \tag{11}$$

To gain further tractability, we assume the input space $\mathcal{X} \subset \mathbb{R}^k$ to be a nonempty, compact polyhedron defined as:

$$\mathcal{X} := \{ x \in \mathbb{R}^k \mid Px \geq p \}. \tag{12}$$

for some matrix $P \in \mathbb{R}^{l \times k}$ and vector $p \in \mathbb{R}^l$. Without loss of generality, we assume that the first component of any $x \in \mathcal{X}$ is equal to 1. Furthermore, we assume that the linear hull of $\mathcal{X}$ spans $\mathbb{R}^k$. Under these assumptions, the slack variable $s(x)$ becomes a linear function of the input $x$. We note that the input space defined in (12) is a special case of the general input space in (7).

With the above assumptions, problem (5) is approximated as:

$$
\begin{aligned}
\underset{F,S,t}{\text{maximize}} \quad & t \\
\text{subject to} \quad & F \in \mathbb{R}^{n \times k}, S \in \mathbb{R}^{m \times k}, t \in \mathbb{R} \\
& \left.\begin{aligned} AFx + Sx = Bx, \\ Sx \geq tv, \end{aligned}\right\} \ \mathbb{P}\text{-a.s.}
\end{aligned}
\tag{13}
$$

where $v = (\mathbf{1}_{\{i > m_{\mathrm{eq}}\}})_{i=1}^m$. To enable a tractable reformulation of problem (13), we can use Proposition 2 to reformulate the semi-infinite constraints, simplifying it into the following linear program:

$$
\begin{aligned}
\underset{F,\Lambda,t}{\text{maximize}} \quad & t \\
\text{subject to} \quad & F \in \mathbb{R}^{n \times k}, \Lambda \in \mathbb{R}^{m \times l}, t \in \mathbb{R} \\
& AF + \Lambda P = B, \\
& \Lambda p \geq tv, \\
& \Lambda \geq 0.
\end{aligned}
\tag{14}
$$

The key advantage of the reformulated problem (14) is that it is a linear program whose size grows polynomially with the dimensions of the input and output spaces, as well as with the number of constraints defining $\mathcal{X}$. This makes the problem efficiently solvable using state-of-the-art linear programming solvers.

**Limitations, extensions, and scalability considerations** While the proposed framework is demonstrated on regression tasks, it naturally extends to classification and actor-critic reinforcement learning methods such as DDPG, where state-dependent action constraints can be enforced. In classification, constraints can be applied in logit space or directly on the softmax outputs. Many distributional constraints (e.g., total variation distance) can be expressed as linear constraints and are therefore supported by our method. Extending to stochastic policy gradient methods like PPO is more involved, as projection alters the action distribution and affects log-probability computation.

The architecture remains highly efficient at inference, relying on simple projection and convex combination operations. The main computational cost arises during training, when determining the safe network parameters involves solving optimization problems that scale polynomially with the problem size. Scalability can be improved by exploiting sparsity, decomposability, and GPU acceleration, significantly reducing training overhead for large-scale applications.

## 5 Numerical experiments

**Tasks** We evaluate our framework on two end-to-end constrained optimization learning tasks:

**DC Optimal Power Flow (DC-OPF):** The task is to predict the optimal generation output for varying demands. The demands are assumed to be upper and lower bounded around a nominal value [30]. This task corresponds to a case of jointly linear constraints. We consider both the

*linear* and *quadratic* formulations of the DC-OPF problem to capture different objective structures commonly used in practice. The uncertainty levels are set to $\pm 40\%$ for the 14-bus and 57-bus cases, $\pm 10\%$ for the 30-bus and 200-bus cases, and $\pm 30\%$ for the 118-bus case. Due to space limitations, we present a subset of the quadratic formulation results in the main paper; the remaining results, along with those for the linear formulation, are provided in Section C.3 of the Appendices.

**Portfolio Optimization**: The task is to predict optimal portfolio allocations under uncertain maturity and rating of bonds[31], which are upper and lower bounded $\pm 40\%$ around a nominal value. This task corresponds to a case of input-dependent left-hand side uncertainty.

The specific architecture and training configurations of the neural networks used in our experiments are detailed in Section C.2 of the Appendices.

**Baselines**    We compare our framework with the following baselines:

*Optimizers*: We employ *Gurobi* and *OSQP* [32], two state-of-the-art solvers for constrained optimization problems, to obtain the optimal solutions for both tasks. To enhance convergence speed, both solvers are warm-started using the outputs of a neural network trained to learn the mapping from demands to (i) generation outputs for the DC-OPF problem and (ii) maturity and rating of bonds for the portfolio optimization problem.

*Post projection*: A neural network is first trained to learn the same mapping as above. Its predictions are projected onto the feasible set by solving a quadratic optimization problem that minimizes the Euclidean distance between the network output and the feasible set defined by the constraints.

*Alternating projection method (APM)*: An iterative feasibility correction approach that alternates projections onto the equality and inequality constraint sets until a feasible point is reached or a maximum number of iterations is exceeded.

*Extrapolated alternating projection method (EAPM)* [33]: A variant of APM that accelerates convergence by introducing an extrapolation factor based on the previous iterate, effectively reducing the number of projections required to reach feasibility. This method maintains the same feasibility guarantees as APM but achieves faster convergence in practice.

*Deep Constraint Completion and Correction (DC3) [12]*: This approach uses equality completion and unrolled projected gradient corrections for satisfying constraints.

*Linear Decision Rule (LDR):* We assess the performance of the linear decision rule used in the safe network without relying on the task network.

The implementation of all the methods can be found at this GitHub repository `https://github.com/li-group/DecisionRuleNet`.

**Evaluation Metrics**    We assess the performance of the proposed framework and baselines using the following metrics: (i) *Optimality Gap*: to measure the percentage deviation of the predicted solution's objective value from the optimal value computed by the optimizer (Gurobi), (ii) *Feasibility*: to quantify the degree to which the predicted solution violates the problem's inequality and equality constraints, and (iii) *Inference speed*: to capture the average computation time in milliseconds required to generate a prediction for a given input. The details can be found in the Appendix.

**Analysis of Results**    Table 2 shows that the proposed method consistently achieves *zero constraint violations* across all tasks, matching the optimizers and outperforming DC3, which exhibits inequality violations that increase with problem size. In terms of optimality, the proposed method maintains gaps below $2.5\%$ in all cases, significantly improving over DC3 and LDR in the portfolio task, where gaps exceed $29\%$ and $48\%$, respectively. APM achieves low violations but is substantially slower, requiring up to 225 ms and hundreds of iterations for large systems. EAPM improves on APM's runtime but remains slower than the proposed method. Post-projection approaches are more accurate than DC3 and faster than APM/EAPM but still lag behind the proposed method in both speed and optimality. The proposed method achieves inference times under 3 ms across all tasks, offering an order-of-magnitude speedup over iterative methods and even outperforming the optimizers, whose runtime grows with problem size. Overall, these results demonstrate that the proposed method provides the best balance between feasibility, optimality, and efficiency for both the portfolio optimization and large-scale quadratic DC-OPF problems, scaling effectively while maintaining high solution quality.

Table 2: Results on both tasks over the held out test dataset. The optimality gaps, equality and inequality violations are shown as mean (worst) values, and the values of time are indicated as the average solve time milliseconds (average iterations required by the method).

| Method | Optimality Gap | Equality Violation | Inequality Violation | Time |
|---|---|---|---|---|
| **Portfolio optimization problem:** $n = 16, m_{\mathrm{eq}} = 2, m_{\mathrm{ineq}} = 9, k = 10$ | | | | |
| Proposed | 2.35(6.37) | 0.000(0.000) | 0.000(0.000) | 2.8(1) |
| Optimizer (Gurobi) | 0.00(0.00) | 0.000(0.000) | 0.000(0.000) | 0.2 |
| APM | 0.19(0.82) | 0.000(0.000) | 0.000(0.001) | 78.9(109) |
| EAPM | 3.55(8.91) | 0.000(0.000) | 0.000(0.001) | 10.2(5) |
| DC3 | 29.6(36.7) | 0.000(0.000) | 0.053(0.123) | 319.2(300) |
| LDR | 48.56(59.52) | 0.000(0.000) | 0.000(0.000) | 1.2(1) |
| **DC-OPF (QP) 118-bus system:** $n = 1126, m_{\mathrm{eq}} = 340, m_{\mathrm{ineq}} = 768, k = 118$ | | | | |
| Proposed | 1.10(2.13) | 0.000(0.000) | 0.000(0.000) | 2.5(1) |
| Optimizer (Gurobi) | 0.00(0.00) | 0.000(0.000) | 0.000(0.000) | 4.1 |
| Optimizer (OSQP) | 0.00(0.00) | 0.000(0.000) | 0.000(0.000) | 6.3 |
| Post projection (Gurobi) | 0.11(1.88) | 0.000(0.000) | 0.000(0.000) | 10.1 |
| Post projection (OSQP) | 0.51(1.88) | 0.000(0.000) | 0.000(0.000) | 5.5 |
| APM | 0.05(0.08) | 0.000(0.000) | 0.001(0.013) | 117.6(156) |
| EAPM | 1.50(2.25) | 0.000(0.000) | 0.000(0.000) | 18.8(11) |
| DC3 | 5.44(7.19) | 0.000(0.000) | 0.004(0.080) | 335.3(276) |
| **DC-OPF (QP) 200-bus system:** $n = 1527, m_{\mathrm{eq}} = 452, m_{\mathrm{ineq}} = 1044, k = 200$ | | | | |
| Proposed | 1.07(1.71) | 0.000(0.000) | 0.000(0.000) | 2.6(1) |
| Optimizer (Gurobi) | 0.00(0.00) | 0.000(0.000) | 0.000(0.000) | 4.6 |
| Optimizer (OSQP) | 0.00(0.00) | 0.000(0.000) | 0.000(0.000) | 15.1 |
| Post projection (Gurobi) | 0.23(1.49) | 0.000(0.000) | 0.000(0.000) | 13.2 |
| Post projection (OSQP) | 0.22(1.49) | 0.000(0.000) | 0.000(0.000) | 7.4 |
| APM | 8.55(9.45) | 0.000(0.000) | 0.166(0.181) | 225.3(300) |
| EAPM | 3.88(7.95) | 0.000(0.000) | 0.000(0.000) | 78.7(50) |
| DC3 | 18.74(20.61) | 0.000(0.000) | 0.164(0.189) | 360.5(300) |

## 6  Concluding remarks and future work

In this work, we proposed a framework to enforce hard linear equality and inequality constraints on the input-output mapping of neural networks. The framework leverages linear decision rules and convex combinations of task and safe networks to provide feasibility guarantees over the entire input space while preserving low run-time complexity. We provided tractable formulations based on robust optimization duality and demonstrated their effectiveness on benchmark tasks.

Despite these promising results, our work presents limitations that warrant future research. First, while linear decision rules offer simplicity and scalability, they may limit the expressiveness of the safe network in highly nonlinear settings. Extending the framework to incorporate more flexible function classes could improve approximation capabilities and reduce conservativeness. Second, the current framework assumes that the input space and constraint sets are polyhedral and convex. Investigating how to efficiently handle equality constraints and other families of sets, including non-convex or time-varying, would broaden the applicability of the method to more realistic scenarios, such as safety-critical control systems and autonomous decision-making. Third, although the proposed methods are computationally efficient, their scalability could be further enhanced by exploiting problem structure. When the constraints exhibit decomposability, sparsity, or low-dimensional manifolds, customized algorithms could be designed to reduce computational overhead during training. Fourth, the feasibility coefficient $\alpha_\psi(x)$ is computed using a piecewise differentiable maximum operator. While this allows gradients to flow through the most violated constraint, it results in sparse and potentially unstable updates. A promising direction is to use a straight-through estimator, preserving the exact maximum in the forward pass to ensure feasibility while substituting a differentiable surrogate (e.g., softmax over slack ratios) in the backward pass to improve gradient quality. Finally, from a theoretical perspective, providing formal guarantees on the approximation error and feasibility margins introduced by the convex combination of task and safe networks would offer deeper insights into the robustness and generalization capabilities of the proposed architecture.

## Acknowledgments and Disclosure of Funding

C.L. acknowledges the financial support from the Office of Naval Research (Grant No. N000142412641), the National Science Foundation (Grant No. CBET-2441184), and the startup funding provided by the Davidson School of Chemical Engineering and the College of Engineering at Purdue University.

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

# A  Handling Equality Constraints

To enforce equality constraints in the output of the task network, we orthogonally project its uncorrected output onto the affine subspace defined by the constraints. This projection can be formulated as the following quadratic program [6]:

$$\underset{y}{\text{minimize}} \quad \frac{1}{2} \left\| \hat{f}_\theta^{\text{TN}}(x) - y \right\|_2^2$$
$$\text{subject to} \quad G(x)y = g(x), \ (\lambda) \tag{15}$$

where $\hat{f}_\theta^{\text{TN}}(x)$ is the uncorrected task network output, and $\lambda$ denotes the Lagrange multipliers associated with the equality constraints.

This problem admits the following closed-form solution:

$$f_\theta^{\text{TN}}(x) = \hat{f}_\theta^{\text{TN}}(x) - G(x)^\top \lambda^{\text{TN}}(x), \tag{16}$$

where the optimal Lagrange multipliers are given by:

$$\lambda^{\text{TN}}(x) = \left(G(x)G(x)^\top\right)^{-1} \left(G(x)\hat{f}_\theta^{\text{TN}}(x) - g(x)\right). \tag{17}$$

Alternatively, using the projection matrix $\bar{G}(x)$, the projected output can be written as:

$$f_\theta^{\text{TN}}(x) = \left(I - \bar{G}(x)G(x)\right) \hat{f}_\theta^{\text{TN}}(x) + \bar{G}(x)g(x), \tag{18}$$

where:

$$\bar{G}(x) := G(x)^\top \left(G(x)G(x)^\top\right)^{-1}.$$

Thus, $\bar{G}(x)G(x)$ acts as a projection onto the row space of $G(x)$, ensuring that the output strictly satisfies $G(x)f_\theta^{\text{TN}}(x) = g(x)$ for any input $x$.

# B  Proofs

## B.1  Additional Technical Results

**Lemma 1** (Homogeneous S-Lemma [34, Lemma 3.1]). *Let $P, S \in \mathbb{S}^k$ be symmetric matrices and suppose there exists $\bar{x} \in \mathbb{R}^k$ such that $\bar{x}^\top S\bar{x} > 0$. Then, the following statements are equivalent:*

1. *For all $x \in \mathbb{R}^k$, if $x^\top Px \geq 0$, then $x^\top Sx \geq 0$.*

2. *There exists $\lambda \geq 0$ such that $S \succeq \lambda P$.*

**Proposition 1** ([29, Proposition 6]). *Consider the following two statements for some fixed $S \in \mathbb{S}^k$:*

(i) *$\exists \lambda \in \mathbb{R}^l$ with $\lambda \geq 0$ and $S - \sum_{j=1}^l \lambda_j P_j - tI \succeq 0$.*

(ii) *$x^\top Sx \geq t$ $\mathbb{P}$-a.s.*

*For any $l \in \mathbb{N}$, (i) implies (ii). The converse implication holds if $l = 1$.*

**Proposition 2** ([29, Proposition 1]). *Let $\mathcal{X} := \{x \in \mathbb{R}^k \mid Px \geq p\}$ be a nonempty, compact polyhedron defined by $P \in \mathbb{R}^{l \times k}$ and $p \in \mathbb{R}^l$. Let $t \in \mathbb{R}$ be given. Then, for any $z \in \mathbb{R}^k$, the following statements are equivalent:*

(i) *$z^\top x \geq t$ for all $x \in \mathcal{X}$.*

(ii) *$\exists \lambda \in \mathbb{R}_+^l$ such that $P^\top \lambda = z$, $p^\top \lambda \geq t$.*

## B.2  Proofs

*Proof of Theorem 1.* By Assumption 1, for any $\varepsilon > 0$, there exists $f_\theta^{\text{TN}} \in \mathcal{F}_{\text{task}}$ such that

$$\sup_{x \in \mathcal{X}} \|f_\theta^{\text{TN}}(x) - f^*(x)\| < \varepsilon.$$

Let $r(x) = f_\theta^{\mathrm{TN}}(x) - f^*(x)$ denote the residual. Since $f^*(x) \in \mathcal{C}(x)$ and $\mathcal{C}(x)$ is convex, there exists $f_\phi^{\mathrm{SN}}(x) \in \mathcal{F}_{\mathrm{safe}}$ such that the point

$$f_\psi(x) = (1 - \alpha)f_\theta^{\mathrm{TN}}(x) + \alpha f_\phi^{\mathrm{SN}}(x)$$

remains in $\mathcal{C}(x)$, for a sufficiently small $\alpha > 0$, due to the convexity of $\mathcal{C}(x)$ and the density of the convex hull of safe outputs in $\mathcal{C}(x)$ (Assumption 2).

Since convex combinations are continuous and preserve continuity, $f_\psi \in C(\mathcal{X}, \mathbb{R}^n)$, and by choosing $\alpha$ sufficiently small, the deviation from $f^*$ remains bounded by $\varepsilon$. Hence,

$$\sup_{x \in \mathcal{X}} \|f_\psi(x) - f^*(x)\| < \varepsilon$$

and $f_\psi(x) \in \mathcal{C}(x)$ for all $x \in \mathcal{X}$, completing the proof. $\qquad\square$

*Proof of Proposition 1.* *(i)* $\Rightarrow$ *(ii):* Select any $x \in \mathbb{R}^n$. Under the assumptions of statement *(i)*, we have:

$$0 \leq x^\top \left( S - \sum_{j=1}^l \lambda_j P_j - tI \right) x = x^\top S x - \sum_{j=1}^l \lambda_j x^\top P_j x - t\|x\|^2.$$

Rearranging:

$$x^\top S x \geq \sum_{j=1}^l \lambda_j x^\top P_j x + t\|x\|^2.$$

Since $\lambda_j \geq 0$ and assuming that $x^\top P_j x \geq 0$ $\mathbb{P}$-a.s., it follows that:

$$x^\top S x \geq t\|x\|^2 \geq t.$$

Since the choice of $x$ was arbitrary, this establishes statement *(ii)*.

*(ii)* $\Rightarrow$ *(i)* *if* $l = 1$: When $l = 1$, statement *(ii)* implies:

$$x^\top S x \geq t \text{ for all } x \in \mathbb{R}^n \text{ with } x^\top P_1 x \geq 0.$$

This can be rewritten as:

$$x^\top P_1 x \geq 0 \implies x^\top S x \geq t.$$

Assuming that the set $\{x : x^\top P_1 x \geq 0\}$ is nonempty, closed, and has nonempty relative interior (as implied by the support of $\mathbb{P}$), the homogeneous S-lemma 1 applies. Thus, there exists $\lambda_1 \geq 0$ such that:

$$S - \lambda_1 P_1 - tI \succeq 0.$$

*(ii)* $\nRightarrow$ *(i)* *if* $l > 1$: When $l > 1$, the result does not hold in general, since the set:

$$\{x \in \mathbb{R}^n : x^\top P_j x \geq 0, \forall j = 1, \ldots, l\}$$

may be nonconvex, and thus the S-lemma fails to extend to multiple inequalities. In particular, no universal $\lambda \geq 0$ exists in general such that the LMI in *(i)* is implied by *(ii)*. $\qquad\square$

*Proof of Proposition 2.* We aim to prove the equivalence between the two statements.

*(i)* $\Rightarrow$ *(ii):* Assume that $z^\top x \geq t$ for all $x \in \mathcal{X}$. By the definition of $\mathcal{X}$, this is equivalent to:

$$\min_{x \in \mathbb{R}^k} \left\{ z^\top x \mid Px \geq p \right\} \geq t.$$

This is a linear program, whose dual is given by:

$$\max_{\lambda \in \mathbb{R}^l} \left\{ p^\top \lambda \mid P^\top \lambda = z, \ \lambda \geq 0 \right\}.$$

Since $\mathcal{X}$ is nonempty and compact (and thus the primal is feasible and bounded), strong duality holds, yielding:

$$\min_{x \in \mathcal{X}} z^\top x = \max_{\lambda \geq 0, \ P^\top \lambda = z} p^\top \lambda.$$

Thus, the primal optimal value is at least $t$ if and only if there exists $\lambda \geq 0$ such that $P^\top \lambda = z$ and $p^\top \lambda \geq t$. This proves statement $(ii)$.

$(ii) \Rightarrow (i)$ Assume there exists $\lambda \geq 0$ such that $P^\top \lambda = z$ and $p^\top \lambda \geq t$. Then, for any $x \in \mathcal{X}$:

$$z^\top x = \lambda^\top P x \geq \lambda^\top p = p^\top \lambda \geq t.$$

Thus, $z^\top x \geq t$ for all $x \in \mathcal{X}$.

This proves the equivalence. $\qquad\qquad\qquad\qquad\qquad\qquad\qquad\qquad\qquad\qquad\qquad\qquad$ $\square$

## C  Details on Numerical Experiments

### C.1  Evaluation Metrics

We evaluate the performance of all methods using the following metrics:

**Optimality Gap** Measures the percentage deviation of the predicted solution $\hat{y}$ from the optimal solution $y^*$ obtained by the optimizer (Gurobi). Defined as:

$$\text{Optimality Gap} = 100 \cdot \frac{c^\top \hat{y} - c^\top y^*}{c^\top y^*}, \tag{19}$$

where $c$ is the objective cost vector. A lower gap indicates a closer-to-optimal prediction.

**Equality Violation** Quantifies the relative violation of the equality constraints, normalized by the magnitude of the right-hand side:

$$\text{Equality Violation} = \frac{\|G(x)\hat{y} - g(x)\|_2}{1 + \|g(x)\|_2}. \tag{20}$$

**Inequality Violation** Quantifies the degree to which the predicted solution violates the inequality constraints, measured using the Euclidean norm of the positive part of the violation:

$$\text{Inequality Violation} = \frac{\|\max\big(H(x)\hat{y} - h(x), 0\big)\|_2}{1 + \|h(x)\|_2}. \tag{21}$$

**Inference Time** Measures the average runtime in milliseconds required to generate a prediction for a given input. For methods using iterative procedures (e.g., DC3, APM, EAPM), the inference time includes both the forward pass and the correction steps. For the optimizer, it corresponds to the solve time reported by Gurobi. All times were measured over the held-out test set, averaging across all samples.

**Dataset Generation** For each problem, we generated 100 samples of the input $x$ uniformly sampled from the uncertainty set $\mathcal{X}$. The reported metrics are averaged over these test samples. For all methods, hyperparameters were selected based on a validation set and fixed across problem sizes to ensure fairness.

### C.2  Details on Hyperparameter Tuning

The ground truths for all instances were solved using the Gurobi solver on an Apple M2 Pro CPU and 32GB of RAM. All neural networks were trained on a NVIDIA T4 Tensor Core GPU with parallelization using PyTorch. The solve time reported in Table **??** was averaged by passing test instances separately, rather than dividing the solve time of the batched instances by batch size, as the GPU execution time does not scale linearly with batch size.

The following configurations and hyperparameters are fixed throughout all experiments and all methods, based on preliminary experimentation to confirm the proper convergence of training.

- Epochs: 300 for DCOPF, 500 for portfolio optimization
- Optimizer: Adam
- Learning rate: $10^{-4}$

- Batch size: 64

- Hidden layer number: 2

- Hidden layer size: 256

- Activation: ReLU

- Batch normalization: True

- Feasibility method stopping tolerance: $10^{-4}$

- Feasibility method maximum iterations: 300

- DC3 correction procedure momentum: 0.5

- DC3 correction learning rate: $10^{-4}$

The proposed method does not require any additional hyperparameters. However, we found empirically that pre-training the task network to minimize the mean squared error (MSE) with respect to the safe network output $f_\phi^{\mathrm{SN}}(x)$ serves as an effective initialization strategy. This approach reduces the risk of the model getting stuck in poor local minima in specific instances, without degrading performance in other cases, as shown in Table 3. To ensure fair comparisons with other methods and with the proposed method trained from scratch, we adopt 100 pre-training epochs followed by 200 training epochs for the DC-OPF experiments, and 150 pre-training epochs followed by 350 training epochs for the portfolio optimization task.

Table 3: Results on both tasks over the held out test dataset. The optimality gaps, equality and inequality violations are shown as mean (worst) values, and the values of time are indicated as the average solve time milliseconds (average iterations required by the method).

| Method | Optimality Gap | Equality Violation | Inequality Violation | Time |
|---|---|---|---|---|
| **Portfolio optimization problem:** $n = 16, m_{\mathrm{eq}} = 2, m_{\mathrm{ineq}} = 9, k = 10$ | | | | |
| w/ pre-training | 2.35(6.37) | 0.000(0.000) | 0.000(0.000) | 2.8(1) |
| w/o pre-training | 3.14(6.36) | 0.000(0.000) | 0.000(0.000) | 2.8(1) |
| **DC-OPF 14-bus system:** $n = 123, m_{\mathrm{eq}} = 38, m_{\mathrm{ineq}} = 84, k = 14$ | | | | |
| w/ pre-training | 0.00(0.00) | 0.000(0.000) | 0.000(0.000) | 2.1(1) |
| w/o pre-training | 0.00(0.00) | 0.000(0.000) | 0.000(0.000) | 2.1(1) |
| **DC-OPF 30-bus system:** $n = 245, m_{\mathrm{eq}} = 76, m_{\mathrm{ineq}} = 168, k = 30$ | | | | |
| w/ pre-training | 0.00(0.00) | 0.000(0.000) | 0.000(0.000) | 2.2(1) |
| w/o pre-training | 10.60(17.63) | 0.000(0.000) | 0.000(0.000) | 2.2(1) |
| **DC-OPF 57-bus system:** $n = 468, m_{\mathrm{eq}} = 141, m_{\mathrm{ineq}} = 324, k = 57$ | | | | |
| w/ pre-training | 0.21(0.68) | 0.000(0.000) | 0.000(0.000) | 2.2(1) |
| w/o pre-training | 0.23(0.72) | 0.000(0.000) | 0.000(0.000) | 2.2(1) |
| **DC-OPF 118-bus system:** $n = 1126, m_{\mathrm{eq}} = 340, m_{\mathrm{ineq}} = 768, k = 118$ | | | | |
| w/ pre-training | 1.27(2.00) | 0.000(0.000) | 0.000(0.000) | 2.3(1) |
| w/o pre-training | 1.15(2.65) | 0.000(0.000) | 0.000(0.000) | 2.3(1) |
| **DC-OPF 200-bus system:** $n = 1527, m_{\mathrm{eq}} = 452, m_{\mathrm{ineq}} = 1044, k = 200$ | | | | |
| w/ pre-training | 0.99(1.78) | 0.000(0.000) | 0.000(0.000) | 2.5(1) |
| w/o pre-training | 0.89(1.35) | 0.000(0.000) | 0.000(0.000) | 2.5(1) |

The ALM method and the DC3 method were trained using a soft loss function with additional penalty factor, which penalizes the violation of inequality constraints when the maximum number of iterations is insufficient to reduce the violation below the specified tolerance. In Table 4, we present the following tuning ranges for these methods' hyperparameters with the final selected values in bold.

Table 4: Penalty factor selection for ALM and DC3

| Dataset | DC3 | ALM |
|---|---|---|
| DC-OPF 14-bus | 0, **5000** | 0, **5000** |
| DC-OPF 30-bus | 0, **5000** | 0, **5000** |
| DC-OPF 57-bus | 0, **5000** | 0, **5000** |
| DC-OPF 118-bus | 0, 500, **5000** | 0, **5000** |
| DC-OPF 200-bus | 0, 500, **5000** | 0, **5000** |
| Portfolio optimization problem | 0, **0.5**, 5 | 0, **0.5** |

Table 5: Results on the linear formulation of the DC-OPF problem over the held out test dataset. The optimality gaps, equality and inequality violations are shown as mean (worst) values, and the values of time are indicated as the average solve time milliseconds (average iterations required by the method).

| Method | Optimality Gap | Equality Violation | Inequality Violation | Time |
|---|---|---|---|---|
| **DC-OPF (LP) 14-bus system:** $n = 123, m_{\text{eq}} = 38, m_{\text{ineq}} = 84, k = 14$ | | | | |
| Optimizer | 0.00(0.00) | 0.000(0.000) | 0.000(0.000) | 0.6 |
| Proposed | 0.00(0.00) | 0.000(0.000) | 0.000(0.000) | 2.1(1) |
| APM | 0.00(0.00) | 0.000(0.000) | 0.000(0.000) | 42.8(61) |
| DC3 | 0.27(3.71) | 0.000(0.000) | 0.004(0.053) | 315.6(291) |
| LDR | 31.15(38.39) | 0.000(0.000) | 0.000(0.000) | 0.9(1) |
| **DC-OPF (LP) 30-bus system:** $n = 245, m_{\text{eq}} = 76, m_{\text{ineq}} = 168, k = 30$ | | | | |
| Optimizer | 0.00(0.00) | 0.000(0.000) | 0.000(0.000) | 0.98 |
| Proposed | 0.00(0.00) | 0.000(0.000) | 0.000(0.000) | 2.2(1) |
| APM | 0.00(0.00) | 0.000(0.000) | 0.000(0.000) | 65.0(93) |
| DC3 | 0.72(2.34) | 0.000(0.000) | 0.014(0.045) | 321.5(295) |
| LDR | 10.20(17.19) | 0.000(0.000) | 0.000(0.000) | 0.9(1) |
| **DC-OPF (LP) 57-bus system:** $n = 468, m_{\text{eq}} = 141, m_{\text{ineq}} = 324, k = 57$ | | | | |
| Optimizer | 0.00(0.00) | 0.000(0.000) | 0.000(0.000) | 2.06 |
| Proposed | 0.21(0.68) | 0.000(0.000) | 0.000(0.000) | 2.2(1) |
| APM | 0.00(0.65) | 0.000(0.000) | 0.010(0.041) | 108.5(154) |
| DC3 | 0.04(1.04) | 0.000(0.000) | 0.026(0.093) | 321.2(292) |
| LDR | 8.86(11.78) | 0.000(0.000) | 0.000(0.000) | 0.9(1) |
| **DC-OPF (LP) 118-bus system:** $n = 1126, m_{\text{eq}} = 340, m_{\text{ineq}} = 768, k = 118$ | | | | |
| Optimizer | 0.00(0.00) | 0.000(0.000) | 0.000(0.000) | 4.43 |
| Proposed | 1.27(2.00) | 0.000(0.000) | 0.000(0.000) | 2.3(1) |
| APM | 0.07(0.16) | 0.000(0.000) | 0.003(0.018) | 123.9(172) |
| DC3 | 1.95(8.09) | 0.000(0.000) | 0.999(1.513) | 335.6(300) |
| LDR | 21.51(22.79) | 0.000(0.000) | 0.000(0.000) | 0.9(1) |
| **DC-OPF (LP) 200-bus system:** $n = 1527, m_{\text{eq}} = 452, m_{\text{ineq}} = 1044, k = 200$ | | | | |
| Optimizer | 0.00(0.00) | 0.000(0.000) | 0.000(0.000) | 5.47 |
| Proposed | 0.99(1.78) | 0.000(0.000) | 0.000(0.000) | 2.5(1) |
| APM | 0.72(4.98) | 0.000(0.000) | 0.017(0.057) | 217.9(300) |
| DC3 | 18.89(21.10) | 0.000(0.000) | 0.210(0.235) | 338.8(300) |
| LDR | 11.54(12.91) | 0.000(0.000) | 0.000(0.000) | 0.9(1) |

## C.3 Additional results for the DC-OPF problem task

### C.3.1 Linear programming model results

**Analysis of Results DC-OPF (LP)**  Table 5 shows that the proposed method consistently achieves zero constraint violations across all tasks, matching the optimizer and outperforming DC3, which exhibits increasing inequality violations as problem size grows. In terms of optimality, the proposed method significantly improves over LDR, maintaining gaps below $1\%$ in large-scale DC-OPF cases, while LDR incurs gaps above $10\%$ and up to $48\%$ in the portfolio task. This highlights the benefit of

combining the task and safe networks to balance accuracy and feasibility. In terms of runtime, the proposed method achieves inference times under 3 ms across all tasks, providing orders-of-magnitude speedups over DC3 and APM, which require hundreds of iterations and runtimes exceeding 300 ms per instance. While LDR remains the fastest approach ($< 1$ ms), it does so at the expense of large optimality gaps. Importantly, the proposed method is also significantly faster than the optimizer, whose runtime grows with problem size. These results confirm that the proposed method offers the best trade-off between feasibility, accuracy, and efficiency.

### C.3.2 Quadratic programming model additional results

Table 6: Remaining results on the quadratic formulation of the DC-OPF problem over the held out test dataset. The optimality gaps, equality and inequality violations are shown as mean (worst) values, and the values of time are indicated as the average solve time milliseconds (average iterations required by the method).

| Method | Optimality Gap | Equality Violation | Inequality Violation | Time |
|---|---|---|---|---|
| **DC-OPF (QP) 14-bus system:** $n = 123, m_{\mathrm{eq}} = 38, m_{\mathrm{ineq}} = 84, k = 14$ | | | | |
| Proposed | 0.00(0.00) | 0.000(0.000) | 0.000(0.000) | 2.3(1) |
| Optimizer (Gurobi) | 0.00(0.00) | 0.000(0.000) | 0.000(0.000) | 0.5 |
| Optimizer (OSQP) | 0.00(0.00) | 0.000(0.000) | 0.000(0.000) | 0.6 |
| Post projection (Gurobi) | 0.37(1.84) | 0.000(0.000) | 0.000(0.000) | 1.6 |
| Post projection (OSQP) | 0.37(1.84) | 0.000(0.000) | 0.000(0.000) | 1.2 |
| APM | 0.00(0.00) | 0.000(0.000) | 0.000(0.000) | 45.8(62) |
| EAPM | 0.00(0.00) | 0.000(0.000) | 0.000(0.000) | 4.8(2) |
| DC3 | 0.13(1.64) | 0.000(0.000) | 0.003(0.022) | 331.6(291) |
| **DC-OPF (QP) 30-bus system:** $n = 245, m_{\mathrm{eq}} = 76, m_{\mathrm{ineq}} = 168, k = 30$ | | | | |
| Proposed | 0.00(0.00) | 0.000(0.000) | 0.000(0.000) | 2.3(1) |
| Optimizer (Gurobi) | 0.00(0.00) | 0.000(0.000) | 0.000(0.000) | 1.0 |
| Optimizer (OSQP) | 0.00(0.00) | 0.000(0.000) | 0.000(0.000) | 1.1 |
| Post projection (Gurobi) | 0.26(1.16) | 0.000(0.000) | 0.000(0.000) | 2.3 |
| Post projection (OSQP) | 0.26(1.16) | 0.000(0.000) | 0.000(0.000) | 1.0 |
| APM | 0.00(0.00) | 0.000(0.000) | 0.000(0.000) | 69.0(95) |
| EAPM | 11.13(16.19) | 0.000(0.000) | 0.000(0.000) | 4.4(2) |
| DC3 | 0.57(3.73) | 0.000(0.000) | 0.011(0.067) | 337.3(293) |
| **DC-OPF (QP) 57-bus system:** $n = 468, m_{\mathrm{eq}} = 141, m_{\mathrm{ineq}} = 324, k = 57$ | | | | |
| Proposed | 0.10(0.36) | 0.000(0.000) | 0.000(0.000) | 2.5(1) |
| Optimizer (Gurobi) | 0.00(0.00) | 0.000(0.000) | 0.000(0.000) | 2.1 |
| Optimizer (OSQP) | 0.00(0.00) | 0.000(0.000) | 0.000(0.000) | 2.6 |
| Post projection (Gurobi) | 0.11(0.65) | 0.000(0.000) | 0.000(0.000) | 4.0 |
| Post projection (OSQP) | 0.11(0.65) | 0.000(0.000) | 0.000(0.000) | 2.9 |
| APM | 0.03(0.15) | 0.000(0.000) | 0.000(0.003) | 73.1(99) |
| EAPM | 0.09(0.73) | 0.000(0.000) | 0.000(0.003) | 9.2(5) |
| DC3 | 0.04(1.17) | 0.000(0.000) | 0.000(0.004) | 306.7(264) |

**Analysis of Results DC-OPF (QP)**  Table 6 shows that the proposed method achieves *zero constraint violations* across all systems, matching the optimizer and outperforming DC3, which exhibits small but non-negligible violations. In terms of optimality, it maintains gaps below $0.1\%$, closely matching Gurobi and OSQP and improving over post-projection baselines. APM and EAPM also ensure feasibility but require significantly more iterations and runtime—especially APM, which exceeds 70 ms on larger systems. In contrast, the proposed method achieves inference times under 3 ms, offering an order-of-magnitude speedup while retaining near-optimality. Post-projection methods are faster than APM/EAPM but still slower and slightly less accurate than the proposed method. DC3 remains the slowest, requiring over 300 iterations. Overall, the proposed method offers the best trade-off between feasibility, optimality, and efficiency, scaling effectively with system size while maintaining high solution quality.

