# OpenReview forum: "Enforcing Hard Linear Constraints in Deep Learning Models with Decision Rules"
_NeurIPS.cc/2025/Conference — NeurIPS 2025 poster_

### Official Review · Reviewer_bPkV · 2025-06-30

**Clarity:** 3
**Significance:** 3
**Originality:** 3
**Rating:** 5
**Confidence:** 4

**Summary:**

This paper proposes to enforce linear constraints on the outputs of ML models by forming the convex combination of the predictions of a pre-trained "task network" with a learned "safe network." The role of the task network is to optimize prediction accuracy, whereas the role of the safe network is to ensure that the final predictions satisfy the problem's underlying constraints. The infinite-dimensional safe network training problem is shown to reduce to either an SDP or an LP, in the special cases where the support of the data distribution is either defined by an intersection of ellipsoids, or by a polytope. Numerical experiments on optimal power flow and portfolio optimization benchmark tasks are conducted, and it is shown that the proposed method yields predictions that always satisfy the desired constraints, while achieving prediction performance on par with state-of-the-art constrained optimization solvers.

**Questions:**

1. Last sentence of Section 3.1: In the notations section, you mentioned that $A_\ell$ is used to represent the $\ell$th row of a matrix $A$, but here, you are using $A_\ell$ to represent a matrix. This might be a little bit confusing, so I'd suggest using consistent notation choices throughout the paper.
2. "Let $s^\text{TN}(x)$ and $s^\text{SN}(x)$ denote the predicted slack variables" These slack variables don't seem to be very well defined. Up until this point, the only slack variables mentioned were in $s(x)$, appearing in (2), to combine the inequality constraints into an equivalent equality constrint. Those slack variables don't appear to be "predicted," but rather they are prespecified by the problem at hand. So, what exactly are these "predicted slack variables"?
3. "Thus, the optimal $\alpha$ can be computed as:" Can you clarify precisely what this $\alpha$ is optimizing?
4. Is there a reason why you formulate the safe network training in the specific form of (5) in which you are essentially maximizing the amount of slack in the constraints? The intuition behind the particular design of the optimization problem (5) would be good to include in the manuscript.
5. Minor formatting issue: In the heading of Section 3.4 (and in other sections), you capitalize the second word, whereas elsewhere in the paper, you only capitalize the first word. It would be best to remain consistent.
6. In Section 3.5, you use the overline notation $\bar{G}$, but you haven't defined it.
7. What is $C_\text{TN}$ in Table 1?
8. "Note that set $\mathcal{X}$ corresponds to the intersection of $l$ ellipsoids..." Wouldn't you need to assume that all $P_j$ are also negative (semi-)definite.
9. Notation in (8): Do you mean to say that $S\in (\mathbb{S}^k)^m$ rather than $S\in\mathbb{S}^m$? Also, why must the slack variable function $s(\cdot)$ be purely quadratic? It seems to me that the assumptions you outlined result in the constraint becoming $s_l(x) = b_l^\top x- x^\top A_l F x$ for $\mathbb{P}$-almost every $x$, indicating that $s_l$ should have both a quadratic and a linear term.
10. Notation in (10): Shouldn't $\Lambda$ also be listed as a decision variable under "maximize"?
11. Proposition 1: I think you mean $S\in\mathbb{S}^k$ again, rather than $S\in\mathbb{S}$.
12. Proposition 1: It would help the reader if you clarified exactly what $W_j$ are here. Are these the same as the $P_j$ matrices in the main body of the paper, i.e., the matrices defining the quadratic constraints of the support set $\mathcal{X}$ over which the semi-infinite quadratic inequality constraints is to hold almost surely? Also, when you employ this result to rewrite (8) into the SDP (10), it seems to me like you are actually tightening the feasible set (i.e., obtaining a lower bound on (8)), assuming that $l\ne 1$ in general. The way the development is currently worded makes seems to suggest that (8) and (10) are equivalent reformulations, which might be misleading if in fact (10) only lower-bounds (8) in general.

**Ethical Concerns:**

["NO or VERY MINOR ethics concerns only"]

**Final Justification:**

The authors have thoroughly responded to my questions and suggestions, including revisions to clarify certain aspects in the manuscript. All of my concerns have been addressed, and therefore I maintain my positive evaluation of the paper.

**Limitations:**

yes

**Quality:**

3

**Strengths And Weaknesses:**

Strengths: The paper is well-written, and proposes an interesting idea. The proposed method is theoretically justified by a universal approximation result with constraint satisfaction guarantees. The reformulation of the paper's primary training problem into finite-dimensional forms is also rigorously justified. The performance in the numerical experiments shows very promising results compared to the baseline methods compared to.

Weaknesses: See my "Questions" below.

---

> ### Author Rebuttal · Authors · 2025-07-31
>
> **Q1**
>
> You are correct that in the notation section, we stated that $ A_\ell $ denotes the $ \ell $th row of a matrix $ A $. However, throughout the main text, we consistently use $ A_\ell $ to represent the $ \ell $th matrix of a set of matrices.
> We have revised the notation section and removed the earlier row-vector definition to accurately reflect the intended meaning of $ A_\ell $ as a matrix rather than a row vector, ensuring consistency throughout the paper.
>
> ---
>
> **Q2**
>
> We thank the reviewer for this insightful comment. The term “predicted slack variables” was not clearly defined in this context. The slack variable $s(x)$ introduced in equation (2) is a modeling construct used to express the input-dependent inequality constraints $H(x) f_\psi(x) \leq h(x)$ in standard form, and it is not learned or predicted by the model.
> In the paragraph following equation (3), the quantities denoted by $ s^{\mathrm{TN}}(x) $ and $ s^{\mathrm{SN}}(x) $ refer instead to *constraint residuals*, that is, the pointwise difference $ h(x) - H(x) f(x) $ that quantifies how much the output of the task or safe network satisfies (or violates) the inequality constraints. These residuals are not slack variables in the optimization sense, nor are they predicted directly by the model.
> To avoid confusion, we removed the term “predicted” in the revised manuscript and rephrase the surrounding text for clarity, as detailed in the next response.
>
> ---
>
> **Q3**
>
> We appreciate the reviewer’s request for clarification. The purpose of $ \alpha $ is not to optimize a task loss or performance metric, but rather to determine the smallest value in the interval $ [0,1] $ such that the convex combination of the task and safe network outputs satisfies the input-dependent inequality constraints.
>
> Formally, for any input $ x \in \mathcal{X} $, we seek the minimal $ \alpha \in [0,1] $ such that $f_\psi(x) = (1 - \alpha) f_\theta^{\mathrm{TN}}(x) + \alpha f_\phi^{\mathrm{SN}} (x)  $
> satisfies $ H(x) f_\psi(x) \leq h(x) $. The equality constraints are already satisfied by construction: the safe network output is feasible by design, and the task network output is projected onto the affine equality set before combination.
>
> To clarify this in the manuscript, we will revise the text as follows:
>
> > **Neural network output**
> > The output of the task network, $ f_\theta^{\mathrm{TN}}(x) $, is projected to satisfy the equality constraints, while the output of the safe network, $ f_\phi^{\mathrm{SN}}(x) $, is constructed to satisfy both equality and inequality constraints for all $ x \in \mathcal{X} $. We define the final network output as in (3). To ensure that the final output satisfies the inequality constraints, we compute the smallest value of $ \alpha_\psi(x) \in [0,1] $ such that
> $$
> H(x) f_\psi(x) \leq h(x),
> $$
> i.e., such that the convex combination is feasible. The equality constraints are satisfied by construction, since both $ f_\theta^{\mathrm{TN}}(x) $ and $ f_\phi^{\mathrm{SN}}(x) $ lie in the affine subspace defined by the equality constraints.
> > To compute $ \alpha_\psi(x) $, we define the residual slack vectors:
> $$
> s_\theta^{\mathrm{TN}}(x) := h(x) - H(x) f_\theta^{\mathrm{TN}}(x), \quad
> s_\phi^{\mathrm{SN}}(x) := h(x) - H(x) f_\phi^{\mathrm{SN}}(x),
> $$
> which quantify the pointwise satisfaction margin of the inequality constraints for each network output.
> > Let $ \mathcal{I} := \\{ i \in \\{1, \ldots, m_{\mathrm{ineq}} \\} \mid s_{\theta,i}^{\mathrm{TN}}(x) < 0 \\} $ denote the set of inequality constraints violated by the task network. For each such constraint, we determine the minimum value of $ \alpha_\psi(x) \in [0,1] $ such that the corresponding convex combination becomes feasible. The tightest such value across all violated constraints yields:
> \begin{equation}
> \alpha_\psi(x) := \max_{i \in \mathcal{I}} \frac{ -s_{\theta,i}^{\mathrm{TN}}(x) }{ s_{\phi,i}^{\mathrm{SN}}(x) - s_{\theta,i}^{\mathrm{TN}}(x) }.
> \end{equation}
> >
> > This construction ensures that the final output $ f_\psi(x) $ satisfies all constraints in $ \mathcal{C}(x) $, while staying as close as possible to the task network output within the feasible set.
>
> ---
>
> **Q4**
>
> The objective in equation (5) is designed to not only ensure feasibility of the safe network output, but also to encourage it to lie deep in the interior of the feasible set by maximizing the total slack in the inequality constraints.
>
> This design choice is motivated by the role of the safe network in the final architecture: since we form a convex combination between the task and safe network outputs to ensure feasibility, a safe network output that lies farther from the boundary provides more "buffer" margin. In particular, it allows a smaller $\alpha$ to be used in equation (4), thereby keeping the final output closer to the task network prediction. If the safe network output were close to the constraint boundary, the convex combination might require a larger $\alpha$, potentially degrading performance.
>
> To clarify this in the manuscript, we will add the following explanation after equation (5):
>
> > The objective in (5) encourages the safe network output to lie deep in the interior of the feasible region. This promotes robustness during the convex combination with the task network: the larger the slacks in the safe output, the more flexibility we have to interpolate while preserving feasibility. As a result, the final output remains closer to the task network prediction, reducing conservativeness.
>
> ---
>
> **Q5**
>
> Thanks for your observation. In the revised version, we have only capitalized the first word of the section headings.
>
> ---
>
> **Q6**
>
> We appreciate the reviewer for catching this omission. The notation $\bar{G}$ in Section 3.5 denotes the left pseudo-inverse of the equality constraint matrix $G \in \mathbb{R}^{m_{\mathrm{eq}} \times n}$, defined as $\bar{G} := G^\top (G G^\top)^{-1}$. This matrix appears in the closed-form projection formula used to enforce the affine equality constraints $Gx = g$, as given in equation (3).
>
> We will revise the text to define $\bar{G}$ explicitly.
>
> ---
>
> **Q7**
>
> We thank the reviewer for pointing this out. $C_{\mathrm{TN}}$ refers to the computational complexity of a single forward pass through the task network. It is used in Table 1 to quantify the inference cost associated with evaluating $f^\mathrm{TN}_\theta(x)$. This is consistent with the description in Section 3.5, where we state that the overall inference complexity is dominated by this operation. We have clarified this in the revised manuscript.
>
> ---
>
> **Q8**
>
> We appreciate the reviewer’s question. However, it is not necessary to assume that the matrices \( P_j \) are negative (semi-)definite. In our setting, the vector \( x \in \mathbb{R}^k \) satisfies \( x_1 = 1 \) almost surely, so the first row and column of each matrix \( P_j \) correspond to this constant term. If we remove the first row and column of \( P_j \), the resulting submatrix (associated with the truly uncertain components) is indeed negative semidefinite. However, our theoretical results do not rely on this structure. Specifically, Lemma 1 and Proposition 1 in our manuscript hold without requiring definiteness of \( P_j \). We will clarify this point in the revised manuscript.
>
> ---
>
> **Q9**
>
> In the manuscript, we make the modeling assumption that $x_1 = 1$ almost surely. This trick allows us to represent any \textit{affine function} of the remaining components $x_{2:}$ (i.e., $x_2, x_3, \ldots, x_k $) as a \textit{linear function} of the full vector $x$. Specifically, any affine function of the form $A x_{2:} + b$ can be expressed as:
> $$A x_{2:} + b = \begin{bmatrix} b & A \end{bmatrix} \begin{bmatrix} 1 \\\\ x_{2:} \end{bmatrix} = \hat{A} x,$$
> where $\hat{A} := \begin{bmatrix} b & A \end{bmatrix} $ and $ x := \begin{bmatrix} 1 \\\\ x_{2:} \end{bmatrix} $.
> Likewise, any \emph{quadratic plus linear} function of $x_{2:}$, such as
> $$
> x_{2:}^\top Q x_{2:} + q^\top x_{2:} + r,
> $$
> can be rewritten as a \emph{symmetric quadratic form} in $x$, $x^\top S x$,
> where $S \in \mathbb{S}^k$ is a symmetric matrix constructed to capture the original $Q, q, r$ terms. This modeling convention simplifies the optimization formulation while preserving generality.
>
> ---
>
> **Q10**
>
> We thank the reviewer for pointing this out. $\Lambda$ is a decision variable in our formulation and should be explicitly listed under the ``maximize'' operator. We have updated the manuscript accordingly.
>
> ---
>
> **Q11**
>
> Thanks for pointing this abuse of notation out. Yes, we meant $S\in\mathbb{S}^k$. We have updated the manuscript accordingly.
>
> ---
>
> **Q12**
>
> We thank the reviewer for pointing out this important clarification. First, regarding the notation: the matrices $W_j$ in Proposition 1 indeed correspond to the matrices $P_j$ introduced in the definition of the support set $\mathcal{X}$ in the main body of the paper. Specifically, each quadratic constraint defining $\mathcal{X}$ is of the form $x^\top P_j x \geq 0$, and these are the same matrices referred to as $W_j$ in Proposition 1. We will update the notation in the appendix to maintain consistency.
>
> Regarding the second point, we agree that Proposition 1 provides a sufficient condition for the satisfaction of a semi-infinite quadratic inequality over a set defined by  $l \geq 1$ quadratic constraints. Therefore, when $l > 1$, the resulting SDP (10) derived from Proposition 1 yields a conservative inner approximation of the feasible set of (8), and hence the optimal value of (10) can be interpreted as a lower bound on the optimal value of (8). However, when $l = 1$, i.e., the support set $\mathcal{X}$ is defined by a single quadratic constraint, equivalence holds. We will revise the manuscript accordingly to clarify that the SDP-based reformulation in (10) is an inner approximation of (8) in the general case and an exact reformulation for $l = 1$.

---

> > ### Comment · Reviewer_bPkV · 2025-08-01
> >
> > I thank the authors for their very thorough responses to my concerns, and their revisions. All of my concerns have been addressed, and I maintain my positive evaluation of the paper.

---

### Official Review · Reviewer_kQGn · 2025-07-01

**Clarity:** 3
**Significance:** 3
**Originality:** 3
**Rating:** 5
**Confidence:** 3

**Summary:**

The paper proposes a new method to enforce linear input-output (i.e. input dependent) constraints for ML models. The key idea is to split the task of prediction subject to constraints into two halves, one half is represented by a task network which tries to output the best prediction, and the other half is represented by a “safe” network which always outputs a feasible point. The key insight is that the “best” (i.e. closest to the task network output) feasible point on the line segment between the task and safe outputs can be calculated in closed form, which is cheaper than e.g. projecting the task network output onto the feasible region. Compared to existing work, the inference and training costs are very low while maintaining good accuracy and zero constraint violations.

**Questions:**

1. What would happen if the constraints are infeasible? (Potentially for specific inputs)

**Ethical Concerns:**

["NO or VERY MINOR ethics concerns only"]

**Final Justification:**

The proposed method is elegant and theoretically sound. The evaluation is thorough, given the limited datasets available, and the proposed method strikes a highly desirable balance between inference speed and solution quality.

**Limitations:**

Yes

**Quality:**

3

**Strengths And Weaknesses:**

## Strengths

1. The method achieves a favorable tradeoff between optimality gap and inference speed. Inference is very fast while producing high quality solutions. Competitors are either 1-2 orders of magnitude slower to achieve better results or are 2-3x faster while achieving significantly worse results.
2. The evaluation is thorough with four baselines, including an exact solution obtained with Gurobi.
3. The paper also includes an approximation theorem showing that the proposed architecture is a universal approximate of constrained functions, which is a neat result.

## Weaknesses

1. Training may be expensive. Training requires solving a semidefinite program to form the safe network. SDPs are harder in general than projection onto the feasible set of linear constraints, so it is not clear if training is more or less expensive than the alternatives.
2. The functional form restrictions in the decision rules formulation may be too strong. It is possible that the linear functional class may not be expressive enough to solve more complex problems, but it is difficult to tell without experimental results. (This is acknowledges in the conclusion)
3. Relatively few datasets are tested. Results are given for portfolio optimization and DC-OPF, but more variety would be ideal. I understand it may be difficult to find good datasets in this area. Perhaps the synthetic instances used in the DC3 paper could be used to give further evidence of the method’s effectiveness?

---

> ### Author Rebuttal · Authors · 2025-07-31
>
> > Training may be expensive...
>
> We thank the reviewer for raising this important concern. We clarify that a semidefinite program (SDP) is only required when the left-hand-side constraint matrices $G(x)$ and $H(x)$ are input-dependent. In this case, the SDP is solved once before training the task network, in order to construct a linear decision rule (LDR). When only the right-hand side $g(x)$ or $ h(x) $ is input-dependent (and the left-hand-side matrices are fixed), the LDR can be determined by solving a single linear program (LP). In both cases, the resulting SDP or LP grows polynomially with the input and output dimensions, and is solved offline, prior to training the task network. While this offline step can be computationally intensive in large-scale settings, it ensures that inference remains fast, deterministic, and constraint-satisfying without the need for runtime optimization. Moreover, this step can be significantly accelerated by exploiting sparsity, decomposable structure in the constraints, and by using modern GPU-accelerated solvers.
>
> > The functional form...
>
> We agree that LDRs may be restrictive in expressive power when modeling highly nonlinear or nonconvex feasible sets. That said, it is important to emphasize that the use of LDRs is confined to the safe network, which acts as a fallback to guarantee feasibility. The task network, by contrast, is a standard deep neural network trained to optimize task performance, and is free to approximate highly expressive mappings. In this way, the overall architecture maintains high capacity while ensuring hard constraint satisfaction through a lightweight convex combination. Despite the conservativeness of the LDRs, our experiments demonstrate that the proposed architecture achieves competitive performance in terms of both accuracy and feasibility across all benchmarks, with optimality gaps often below 1\% and zero constraint violations.
>
> > Relatively few datasets...
>
> We thank the reviewer for this valuable suggestion. We agree that broader empirical coverage is important. In the revised version of the manuscript, we have significantly extended our experimental evaluation. In particular, we now include: (i) new results for DC-OPF with a quadratic objective function; (ii) comparisons against commercial and open-source optimizers (Gurobi and OSQP), both warm-started with a trained unconstrained neural network; (iii) post-projection of unconstrained neural network outputs onto the feasible set using the same warm starts; and (iv) an implementation of EAPM, a state-of-the-art baseline for hard constraint enforcement. These additions offer a comprehensive and rigorous assessment of the proposed method’s performance and scalability. Regarding the DC3 synthetic instances, we appreciate the suggestion and carefully examined their suitability. However, as described in Section 4.1 of the DC3 paper, these problems are constructed such that any variable projected onto the equality constraints automatically satisfies the inequality constraints. As a result, the safe network becomes inactive during training and the projection architecture reduces to a degenerate case. This makes those benchmarks ineffective for testing the benefits of our full framework, especially the safe network’s role in ensuring strict feasibility under inequality constraints.
> The results of the new experiments are presented at the end of this rebuttal.
>
> ## Questions
>
> **Q1**
>
> We thank the reviewer for raising this important point. Our approach assumes that the linear constraint set is feasible for all $x \in \mathcal{X}$. This is explicitly addressed during the formulation of the safe network, where the parameters of the linear decision rule are obtained by solving an optimization problem, either an SDP or LP, depending on the structure of the constraints, that ensures feasibility holds almost surely over the input domain $\mathcal{X}$. If, for some reason, the constraints are infeasible for a given input within the input space, then the finite-dimensional reformulations used to compute the linear decision rule (problems (10) or (14)) become infeasible, and no valid parameters can be found. In such cases, one practical remedy is to compute a conservative inner approximation of the input space such that the constraints are feasible for all $x \in \mathcal{X}$. Alternatively, one could partition the input space and learn separate linear decision rules for each region, ensuring feasibility holds locally within each partition. These strategies allow the method to retain feasibility guarantees even when global feasibility is not possible by a single linear decision rule.
>
>
> ### Results on both tasks over the held out test dataset
>
> #### DC-OPF (QP) 14-bus system:
>
> | Method               | Optimality Gap | Equality Violation | Inequality Violation | Time       |
> |----------------------|----------------|---------------------|-----------------------|------------|
> | Proposed             | 0.00 (0.00)    | 0.000 (0.000)       | 0.000 (0.000)         | 2.3 (1)    |
> | Optimizer (Gurobi)   | 0.00 (0.00)    | 0.000 (0.000)       | 0.000 (0.000)         | 0.5        |
> | Optimizer (OSQP)     | 0.00 (0.00)    | 0.000 (0.000)       | 0.000 (0.000)         | 0.6        |
> | Post proj. (Gurobi)  | 0.37 (1.84)    | 0.000 (0.000)       | 0.000 (0.000)         | 1.6        |
> | Post proj. (OSQP)    | 0.37 (1.84)    | 0.000 (0.000)       | 0.000 (0.000)         | 1.2        |
> | APM                  | 0.00 (0.00)    | 0.000 (0.000)       | 0.000 (0.000)         | 45.8 (62)  |
> | EAPM                 | 0.00 (0.00)    | 0.000 (0.000)       | 0.000 (0.000)         | 4.8 (2)    |
> | DC3                  | 0.13 (1.64)    | 0.000 (0.000)       | 0.003 (0.022)         | 331.6 (291)|
>
> #### DC-OPF (QP) 30-bus system:
>
> | Method               | Optimality Gap | Equality Violation | Inequality Violation | Time       |
> |----------------------|----------------|---------------------|-----------------------|------------|
> | Proposed             | 0.00 (0.00)    | 0.000 (0.000)       | 0.000 (0.000)         | 2.3 (1)    |
> | Optimizer (Gurobi)   | 0.00 (0.00)    | 0.000 (0.000)       | 0.000 (0.000)         | 1.0        |
> | Optimizer (OSQP)     | 0.00 (0.00)    | 0.000 (0.000)       | 0.000 (0.000)         | 1.1        |
> | Post proj. (Gurobi)  | 0.26 (1.16)    | 0.000 (0.000)       | 0.000 (0.000)         | 2.3        |
> | Post proj. (OSQP)    | 0.26 (1.16)    | 0.000 (0.000)       | 0.000 (0.000)         | 1.0        |
> | APM                  | 0.00 (0.00)    | 0.000 (0.000)       | 0.000 (0.000)         | 69.0 (95)  |
> | EAPM                 | 11.13 (16.19)  | 0.000 (0.000)       | 0.000 (0.000)         | 4.4 (2)    |
> | DC3                  | 0.57 (3.73)    | 0.000 (0.000)       | 0.011 (0.067)         | 337.3 (293)|
>
> #### DC-OPF (QP) 57-bus system:
>
> | Method               | Optimality Gap | Equality Violation | Inequality Violation | Time       |
> |----------------------|----------------|---------------------|-----------------------|------------|
> | Proposed             | 0.10 (0.36)    | 0.000 (0.000)       | 0.000 (0.000)         | 2.5 (1)    |
> | Optimizer (Gurobi)   | 0.00 (0.00)    | 0.000 (0.000)       | 0.000 (0.000)         | 2.1        |
> | Optimizer (OSQP)     | 0.00 (0.00)    | 0.000 (0.000)       | 0.000 (0.000)         | 2.6        |
> | Post proj. (Gurobi)  | 0.11 (0.65)    | 0.000 (0.000)       | 0.000 (0.000)         | 4.0        |
> | Post proj. (OSQP)    | 0.11 (0.65)    | 0.000 (0.000)       | 0.000 (0.000)         | 2.9        |
> | APM                  | 0.03 (0.15)    | 0.000 (0.000)       | 0.000 (0.003)         | 73.1 (99)  |
> | EAPM                 | 0.09 (0.73)    | 0.000 (0.000)       | 0.000 (0.003)         | 9.2 (5)    |
> | DC3                  | 0.04 (1.17)    | 0.000 (0.000)       | 0.000 (0.004)         | 306.7 (264)|
>
> #### DC-OPF (QP) 118-bus system:
>
> | Method               | Optimality Gap | Equality Violation | Inequality Violation | Time       |
> |----------------------|----------------|---------------------|-----------------------|------------|
> | Proposed             | 1.10 (2.13)    | 0.000 (0.000)       | 0.000 (0.000)         | 2.5 (1)    |
> | Optimizer (Gurobi)   | 0.00 (0.00)    | 0.000 (0.000)       | 0.000 (0.000)         | 4.1        |
> | Optimizer (OSQP)     | 0.00 (0.00)    | 0.000 (0.000)       | 0.000 (0.000)         | 6.3        |
> | Post proj. (Gurobi)  | 0.11 (1.88)    | 0.000 (0.000)       | 0.000 (0.000)         | 10.1       |
> | Post proj. (OSQP)    | 0.51 (1.88)    | 0.000 (0.000)       | 0.000 (0.000)         | 5.5        |
> | APM                  | 0.05 (0.08)    | 0.000 (0.000)       | 0.001 (0.013)         | 117.6 (156)|
> | EAPM                 | 1.50 (2.25)    | 0.000 (0.000)       | 0.000 (0.000)         | 18.8 (11)  |
> | DC3                  | 5.44 (7.19)    | 0.000 (0.000)       | 0.004 (0.080)         | 335.3 (276)|
>
> #### DC-OPF (QP) 200-bus system:
>
> | Method               | Optimality Gap | Equality Violation | Inequality Violation | Time       |
> |----------------------|----------------|---------------------|-----------------------|------------|
> | Proposed             | 1.07 (1.71)    | 0.000 (0.000)       | 0.000 (0.000)         | 2.6 (1)    |
> | Optimizer (Gurobi)   | 0.00 (0.00)    | 0.000 (0.000)       | 0.000 (0.000)         | 4.6        |
> | Optimizer (OSQP)     | 0.00 (0.00)    | 0.000 (0.000)       | 0.000 (0.000)         | 15.1       |
> | Post proj. (Gurobi)  | 0.23 (1.49)    | 0.000 (0.000)       | 0.000 (0.000)         | 13.2       |
> | Post proj. (OSQP)    | 0.22 (1.49)    | 0.000 (0.000)       | 0.000 (0.000)         | 7.4        |
> | APM                  | 8.55 (9.45)    | 0.000 (0.000)       | 0.166 (0.181)         | 225.3 (300)|
> | EAPM                 | 3.88 (7.95)    | 0.000 (0.000)       | 0.000 (0.000)         | 78.7 (50)  |
> | DC3                  | 18.74 (20.61)  | 0.000 (0.000)       | 0.164 (0.189)         | 360.5 (300)|

---

> > ### Comment · Reviewer_kQGn · 2025-08-04
> >
> > I would like to thank the authors for their response. They have addressed all of my concerns, so I am raising my score to a 5.

---

### Official Review · Reviewer_nG53 · 2025-07-03

**Clarity:** 3
**Significance:** 3
**Originality:** 3
**Rating:** 5
**Confidence:** 4

**Summary:**

This paper proposes a framework to enforce linear equality and inequality constraints in neural networks. It combines a task network for accuracy with a safe network trained using optimization principles, producing feasible predictions through convex combination.

For the task network, the authors use the Euclidean projection to project the neural network outputs onto the region of linear constraints. Since only linear constraints are involved in this part, closed-form expressions can be derived. In addition, the authors reformulate the training of the safe network as an optimization problem and reformulate the corresponding problem as an SDP or LP when the safe network are confined to linear functions.

Case studies on DCOPF and portfolio optimization demonstrate certain advantages of the proposed method over some existing approaches.

**Questions:**

Q1. Separating the constraint set into equality and inequality subsets for individual treatment is a common approach in optimization algorithms. For example, OSQP employs the Alternating Direction Method of Multipliers (ADMM), where equality constraints are transformed into a quadratic equality-constrained problem to obtain a closed-form solution (similar to the method described in Appendix A of the paper). Since this step involves solving a complex system of linear equations, both approaches have a computational complexity of $O(m^3)$. For inequality constraints, OSQP uses Euclidean projection. Through alternating iterations between these two steps, OSQP gradually converges from an initial point toward feasibility and optimality. The reviewer wants to point out that the APM used in the numerical experiments in Section 5 appears to differ significantly from ADMM as used in OSQP, because ADMM also includes additional Lagrangian multipliers and quadratic penalty terms that enhance convergence performance. Therefore, it is possible that the authors may have employed a relatively weak baseline.

Q2. Considering that a linear system with complexity $O(m^3)$ needs to be solved anyway, instead of using the proposed method, would it be faster to use an unconstrained neural network to predict an initial value and then perform a warm start in Gurobi Simplex by setting its `PStart` or in OSQP by setting its initial values?

Q3. DCOPF on small-scale systems can be easily solved by commercial solvers, so applying the proposed method offers limited value, especially considering that the method may also introduce optimality gaps. How does the proposed method perform on large-scale systems, such as power grids with several thousands or even ten thousand buses?

Q4. The solution times for APM and DC3 shown in Table 2 are quite long, which seems difficult to understand for a small-scale DCOPF problem. Have the authors compared with other optimization-based constrained learning methods, such as CvxpyLayers or OptNet?

Q5. Based on Eq. (4), it seems that $\alpha$ is related to $s_i^{\rm{TN}}(x)$ and $s_i^{\rm{SN}}(x)$, and $s_i^{\rm{TN}}(x)$ and $s_i^{\rm{SN}}(x)$ are related to $\theta$ and $\phi$, namely the parameters of the task network and safe network. Should $\alpha$ be denoted as $\alpha(\theta,\phi)$ and participate in backpropagation? It seems that currently $\alpha$ participates in backpropagation according to `alpha = torch.max(masked_alphas, dim=-1).values` in line 331 of `models.py`.

**Ethical Concerns:**

["NO or VERY MINOR ethics concerns only"]

**Final Justification:**

The authors' response has addressed my concerns, and therefore I have increased my score.

**Limitations:**

Yes.

**Paper Formatting Concerns:**

No concerns.

**Quality:**

3

**Strengths And Weaknesses:**

Strengths:
The authors present their ideas clearly, with well-written content and detailed mathematical derivations.

Weaknesses:
The authors used a small-scale case study and compared with weak baselines, making it difficult to fully demonstrate the advantages of the proposed method. The results in Table 2 do not show an order-of-magnitude improvement in solution time for the proposed method compared to Gurobi. It seems that using the proposed method does not provide much value.

---

> ### Author Rebuttal · Authors · 2025-07-31
>
> **Comment**
> > The authors used a small-scale case...
>
> **Response**
> We appreciate the reviewer’s concerns. In response, we have expanded the experiments to include stronger baselines: (i) Gurobi and OSQP warm-started with unconstrained neural network outputs, (ii) projection-based methods using QP solvers, and (iii) the extended alternating projection method (EAPM), a state-of-the-art iterative approach.
>
> While runtime gains may seem modest on small problems, our method enables millisecond-level, one-shot inference with guaranteed constraint satisfaction, avoiding iterative overhead and making it well-suited for real-time and embedded applications. We also evaluate problems with up to 1,527 variables and 1,496 constraints, sizes comparable to or larger than those in related work, such as [3], [5], [6], [8], [10], [12], [15], [17], [18], and [21], demonstrating the method’s practicality for domains like robotics, power systems, and constrained control.
>
>
> ### Questions
>
> **Q1**
>
> We thank the reviewer for highlighting the distinction between the APM used in our original experiments and ADMM-based solvers such as OSQP. We agree that ADMM introduces dual variables and penalty terms that can enhance convergence properties. However, our proposed method is not an iterative solver and does not rely on convergence guarantees. Instead, it performs one-shot inference via a convex combination of two networks, ensuring feasibility by construction.
>
> To strengthen our evaluation, we have added experiments comparing against a wider set of baselines. These include (1) solving the original optimization problems using Gurobi and OSQP with warm-starts from a trained unconstrained neural network, (2) training an unconstrained neural network and projecting its outputs onto the feasible set via a QP solved using Gurobi and OSQP, and (3) the state-of-the-art Extrapolated Alternating Projection Method (EAPM). As shown in the new experiments, our proposed approach consistently achieves zero constraint violations and competitive optimality gaps while maintaining millisecond-level inference times. It outperforms NN + post-projection methods in both optimality and speed, and achieves comparable or faster inference than OSQP and Gurobi for systems with hundreds to over one thousand variables. These results confirm that our architecture offers a practical, efficient, and constraint-satisfying alternative to traditional and projection-based baselines.
>
> **Q2**
>
> We thank the reviewer for the suggestion. We implemented the proposed warm-start baseline by training an unconstrained neural network (NN) to predict a solution, which is then projected using Gurobi or OSQP, warm-started with that prediction. We also compared against directly solving the original optimization problem with warm starts. Due to space limitations, the results for the largest instance are included at the end of this rebuttal, whereas the full results of the new experiments are included in our response to Reviewer `kQGn`.
>
> The key findings of the numerical experiments are as follows:
>
> * Our proposed method achieves low optimality gaps across all cases, with an almost-fixed-cost inference time (1 forward pass and one convex combination) of 2-3 ms per instance.
> * Warm-starting solvers with a neural network prediction followed by post-projection yields feasible solutions, but results in higher optimality gaps (e.g., up to 1.88 in the 118-bus case), and incurs inference times that increase with system size (e.g., up to 13.2 ms with Gurobi in the 200-bus case versus 2.6 ms incurred by the proposed approach).
> * Solving the original QP with warm-started Gurobi or OSQP does achieve optimality, but comes at the cost of slower inference.
>
> Finally, we emphasize that the cost of solving a linear system arises *only if* the equality constraint matrix \$G(x)\$ is input-dependent. In the common case where \$G\$ is fixed, we compute the projection matrix \$\bar{G}\$ once before training, and reuse it throughout, further reducing the inference complexity.
>
> **Q3**
>
>
> We thank the reviewer for this question. Our method is not intended to replace optimization solvers in centralized or offline workflows, but rather to enable fast and feasible inference in settings where runtime solver calls are impractical.
>
> The benefits of our approach are particularly pronounced when the objective is nonconvex and global solvers are intractable. For example, in the nonconvex, linearly constrained problem included in our new experiments, our method achieves a 6x speedup over Ipopt even though the problem involves only 100 variables, 50 equality constraints, and 50 inequality constraints. While commercial solvers efficiently solve small linear programs, our method demonstrates superior speed in finding feasible solutions for problems with over 1,000 variables, and its advantage becomes increasingly significant as problem size grows.
>
> We also evaluate our method on instances involving over 1,500 variables and nearly 1,500 constraints. Our approach consistently achieves zero constraint violations and inference times in the millisecond range. These problem sizes are comparable to or exceed those studied in prior work, including \[3], \[5], \[6], \[8], \[10], \[12], \[15], \[17], \[18], and \[21], and are representative of real-world applications in power systems, robotics, and constrained control.
>
> The primary scalability challenge lies in determining the parameters of the safe network. We explicitly acknowledge this in our conclusion and propose several strategies to address it, including exploiting sparsity in the constraint matrices, leveraging decomposable constraint structure, and utilizing GPU-accelerated solvers for scalable LDR training.
>
> Lastly, we would like to note that although our case studies focus on parametric LPs, the proposed method is broadly applicable to any setting where linear constraints need to be enforced in deep learning models. For example, it can be used in reinforcement learning problems to ensure that state-action pairs satisfy specified linear constraints, or in applications where a deep learning model approximates an expensive simulator while maintaining feasibility with respect to linear constraints. In these cases, since the problem itself is not an optimization problem, solvers like Gurobi are not applicable. Our method provides a significantly faster alternative to existing approaches for enforcing linear constraints in neural networks, as demonstrated by our experimental results. We have left these additional case studies for future work.
>
> **Q4**
>
> We thank the reviewer for this constructive suggestion. We did experiment with `Cvxpylayers`, but we observed several issues that made it unsuitable for inclusion in the final results. In particular, the inference times were significantly higher than other approaches, often by one to two orders of magnitude. Moreover, `Cvxpylayers` does not strictly enforce inequality constraints during inference, due to the relatively high default solver tolerances. Attempts to tighten these tolerances often led to numerical instability or solver failures, especially for larger instances. For these reasons, we opted not to include `Cvxpylayers` in our results.
> However, we have incorporated several strong optimization-based baselines as suggested. These include training an unconstrained neural network followed by post-projection onto the linear constraint set using state-of-the-art QP solvers such as OSQP and Gurobi (with warm starts), as well as EAPM, a recent variant of alternating projections designed for improved convergence. These methods are described in detail in the revised manuscript and evaluated comprehensively in the new experiments.
>
> **Q5**
>
> We thank the reviewer for raising this insightful question. As noted, $\alpha(x) $ is computed from the slack residuals $s_i^{\mathrm{TN}}(x)$ and $s_i^{\mathrm{SN}}(x)$, which depend on the outputs of the task and safe networks parameterized by $\theta$ and $\phi$, respectively. Therefore, $\alpha(x)$ is indeed a function of these parameters. In this case, gradients do propagate through $\alpha(x)$ in our implementation: PyTorch's `torch.max` operation supports automatic differentiation, and as confirmed by our computational graph, backpropagation flows through the selected (i.e., maximal) index in the slack ratio expression. This gradient, however, is sparse and non-smooth—only the constraint that achieves the maximum receives a nonzero gradient, while all others are effectively ignored. Consequently, the signal reaching the task network is limited in expressiveness, especially when many constraints are near-active. To clarify this in the manuscript, we have updated the notation of $\alpha$ and the slack vectors to reflect their dependence on the network parameters, and we will add a clarifying paragraph in Section 3.2 and Section 6.
>
> ### Results over the held out test dataset: DC-OPF (QP) 200-bus system
>
> | Method               | Optimality Gap | Equality Violation | Inequality Violation | Time       |
> |----------------------|----------------|---------------------|-----------------------|------------|
> | Proposed             | 1.07 (1.71)    | 0.000 (0.000)       | 0.000 (0.000)         | 2.6 (1)    |
> | Optimizer (Gurobi)   | 0.00 (0.00)    | 0.000 (0.000)       | 0.000 (0.000)         | 4.6        |
> | Optimizer (OSQP)     | 0.00 (0.00)    | 0.000 (0.000)       | 0.000 (0.000)         | 15.1       |
> | Post proj. (Gurobi)  | 0.23 (1.49)    | 0.000 (0.000)       | 0.000 (0.000)         | 13.2       |
> | Post proj. (OSQP)    | 0.22 (1.49)    | 0.000 (0.000)       | 0.000 (0.000)         | 7.4        |
> | APM                  | 8.55 (9.45)    | 0.000 (0.000)       | 0.166 (0.181)         | 225.3 (300)|
> | EAPM                 | 3.88 (7.95)    | 0.000 (0.000)       | 0.000 (0.000)         | 78.7 (50)  |
> | DC3                  | 18.74 (20.61)  | 0.000 (0.000)       | 0.164 (0.189)         | 360.5 (300)|

---

> > ### Author Response · Authors · 2025-08-05
> >
> > We would like to thank the reviewer for the constructive comments. As the discussion deadline approaches, we kindly ask if the reviewer has any further questions or concerns.

---

> > ### Comment · Reviewer_nG53 · 2025-08-05
> > **Official Comment by Reviewer nG53**
> >
> > I would like to thank the authors for their response. The authors' response has adequately addressed most of my concerns, and therefore I have increased my score.
> >
> > Besides, I still want to add some explanations about Q2. The authors might have some misunderstandings regarding the method I suggested for comparison. What I proposed was to directly set the initial point for the original optimization problem (for example, directly setting PStart in Gurobi and then solving the original linear programming problem, thereby ensuring the optimal solution can be obtained), rather than using the post-projection method which minimizes the distance to feasible region and thus results in an optimality gap. Nevertheless, since post-projection is a widely used approach, it is acceptable for the author to compare against it.

---

> > > ### Author Response · Authors · 2025-08-05
> > >
> > > We thank the reviewer for the clarification request. We would like to clarify that we correctly implemented the approach suggested by the reviewer: the results labeled "Optimizer (Gurobi)" and "Optimizer (OSQP)" in the response to Reviewer `kQGn` correspond to solving the original optimization problem using warm-starts directly provided by the unconstrained neural network. We note that among the two solvers, OSQP benefits the most from using warm-starts in terms of solution time. We appreciate the reviewer’s confirmation and the increased score.

---

> > > > ### Comment · Reviewer_nG53 · 2025-08-05
> > > > **Official Comment by Reviewer nG53**
> > > >
> > > > Thank you for your clarification. Due to the abbreviation used in the table, I initially misunderstood that the reported time for Optimizer (Gurobi) and Optimizer (OSQP) was for solving OPF problems while directly using the solver's default initial values. I have accordingly adjusted my score, and I have no further questions.

---

### Official Review · Reviewer_Xhps · 2025-07-03

**Clarity:** 4
**Significance:** 4
**Originality:** 4
**Rating:** 5
**Confidence:** 2

**Summary:**

This paper presents a framework to enforce linear input-output constraints during neural neural network training. The idea is to use two separate components: the task network, trained to minimize prediction loss, and a "safe" network, designed to ensure feasibility of the constraints. The output is a convex combination of the two. The approach is evaluated on benchmark regression tasks.

**Questions:**

Please comment on the sizes of the neural network.

How woudl the approach apply to ACAS-Xu (https://arxiv.org/abs/1702.01135) which has a set of input-output constraints.

**Ethical Concerns:**

["NO or VERY MINOR ethics concerns only"]

**Quality:**

4

**Strengths And Weaknesses:**

Pros:

* Interesting paper that achieves safety constraints at a seemingly low cost.

* Interesting theoretical analysis of the properties of the framework.

* Practical evaluation on regression tasks.

Cons:

* Although the authors claim scalability, it is unclear if the approach would scale to large networks and large input domains. Also it is unclear if it extends beyond regression.

* It is unclear if the proposed approach does indeed guarantee absence of violations of constraints.

---

> ### Author Rebuttal · Authors · 2025-07-31
>
> ### Comments on Weaknesses
>
> > **Reviewer Comment**
> > Although the authors claim scalability, it is unclear if the approach would scale to large networks and large input domains. Also it is unclear if it extends beyond regression.
>
> **Response**
> **Scalability.** Our architecture is scalable at inference time, as constraint enforcement relies only on efficient matrix-vector operations, specifically, projection and convex combination steps, as shown in Section 3.5 and Table 1. The main computational cost arises in determining the weights of the safe network, which must satisfy hard input-dependent constraints over $\mathcal{X}$. However, the size of the associated optimization problems scales *polynomially* with the dimensions of the input and output spaces, ensuring tractability in principle.
> To further improve scalability in practice, we outline in Section 5 several promising directions, including exploiting sparsity in the constraint matrices and leveraging decomposable or separable structure in the constraints. Additionally, solving large-scale safe network optimization problems could benefit from modern GPU-based solvers, which have demonstrated significant speedups for solving exa-scale optimization problems. We will clarify in the manuscript that the scalability challenge lies primarily in determining the parameters of the safe network, and that both algorithmic structure and hardware acceleration can mitigate this limitation.
>
> **Beyond regression.** While the current experiments focus on regression tasks, the proposed framework naturally extends to both classification and reinforcement learning (RL) settings.
> In classification settings, the proposed architecture can be applied either in logit space or by imposing constraints directly on the softmax outputs. Constraints involving probability distributions or statistical distances, such as total variation distance, can often be expressed as linear constraints and are therefore naturally compatible with our framework.
> In reinforcement learning, the proposed architecture is directly applicable to enforcing hard, state-dependent constraints on the action space. In actor-critic algorithms such as DDPG, where the policy is deterministic and outputs a continuous action, the projection and blending architecture can be incorporated into the actor network with minimal overhead, as long as the state space is representable in the structured form described in our paper (e.g., a polytope or ellipsoidal set). This allows the agent to output actions that satisfy state-dependent linear constraints exactly at every time step.
> For stochastic policy gradient methods such as PPO, incorporating our architecture is more involved. In particular, the projection step alters the support and shape of the action distribution, making it nontrivial to compute log probabilities required for policy optimization and entropy regularization. Addressing this issue would require adapting the policy parametrization or defining a constrained distribution class over the feasible set—an important and challenging direction for future work.
>
> To summarize these points clearly in the paper, we will add the following to the end of Section 4:
>
> > *Limitations and extensions.* While the proposed framework is demonstrated on regression tasks, it extends naturally to classification (e.g., via constrained logits or softmax outputs) and actor-critic reinforcement learning algorithms such as DDPG, where constrained actions can be enforced in a state-dependent manner. In classification settings, the architecture can operate in logit space or enforce constraints directly on the softmax outputs. Constraints involving distributions or statistical distances, such as total variation distance, can often be written as linear constraints and are therefore naturally supported by our method. Extension to stochastic policy gradient methods like PPO is more involved, as the projection step alters the action distribution and complicates log-probability computation. Although solving the safe network problem scales polynomially with the problem size, its solution can be made more efficient and scalable by exploiting sparsity, decomposability, and GPU-based optimization solvers.
>
> ---
>
> > **Reviewer Comment**
> > It is unclear if the proposed approach does indeed guarantee absence of violations of constraints.
>
> **Response**
> We appreciate the reviewer for raising this important point. Yes, the proposed method guarantees strict satisfaction of all input-dependent constraints—both equalities and inequalities—at training and inference time, by construction.
>
> **Equality constraints** are enforced through a closed-form projection of the task network output onto the affine subspace defined by $G(x) f = g(x)$, as described in equation (3). This projection ensures that the task network output satisfies all equality constraints exactly for any input $x \in \mathcal{X}$. Moreover, by construction, the safe network output satisfies both equality and inequality constraints for all $x \in \mathcal{X}$. As a result, the outputs of both networks, as well as any convex combination between them, lie in the affine subspace defined by the equality constraints $G(x) f = g(x)$.
>
> **Inequality constraints** are enforced via a convex combination of the projected task network output and the output of a safe network that is feasible by design. The interpolation parameter $\alpha$ is computed using equation (4) to be the smallest value in $[0,1]$ that ensures all inequalities $H(x) f(x) \leq h(x)$ are satisfied. This follows directly from the convexity of the constraint set $\mathcal{C}(x)$ and the fact that both endpoints of the interpolation lie within or outside it in a controlled way.
> We have clarified this point more explicitly in Section 3.2 to make the guarantee of hard constraint satisfaction more prominent.
>
> ---
>
> ### Questions
>
> > **Reviewer Question**
> > Please comment on the sizes of the neural network.
>
> **Response**
> Thank you for the suggestion. In the supplementary material (see Section C.2), we provided the architecture details, including the number of layers, hidden units, and activation functions. We chose this configuration based on validation performance. If the paper is accepted, we will include a brief summary of the architecture in the main paper to improve clarity.
>
> ---
>
> > **Reviewer Question**
> > How would the approach apply to ACAS-Xu ([https://arxiv.org/abs/1702.01135](https://arxiv.org/abs/1702.01135)) which has a set of input-output constraints?
>
> **Response**
> We thank the reviewer for highlighting the ACAS Xu case study. While this benchmark provides a compelling application of constrained neural networks, our approach is not directly applicable to the properties $\phi_1$–$\phi_{10}$ described in the Reluplex paper. These properties are formulated as logical implications, i.e., "if the input lies within a certain region, then a desired property must hold for the output", which cannot be encoded as direct linear constraints on inputs and outputs.
> In contrast, our method is designed to enforce linear equalities and inequalities on network outputs, either unconditionally or over convex subsets of the input space. Therefore, while we can enforce fixed output preferences in specific regions, we cannot capture implication-based properties without resorting to additional logic-based mechanisms.

---

> > ### Comment · Reviewer_Xhps · 2025-08-04
> >
> > Thank you for the answers. I maintain my score.

---

### Decision · Program_Chairs · 2025-09-17

**Decision:**

Accept (poster)

**Comment:**

The paper provides a method for adding input-output constraints on networks. This is achieved by splitting the task in between two networks, the main network that aims to reduce the loss, and a safe network that imposes the constraints. The main concerns raised by the reviewers involve the scalability of the method, which was addressed by the reviewers during the rebuttal period. I recommend acceptance, but suggest adding notes on scalability and the cost of solving SDPs.